



# Forest Fire Aerosol – Weather Feedbacks over Western North America Using a High-Resolution, Fully Coupled, Air-Quality Model

Paul A. Makar[1], Ayodeji Akingunola[1], Jack Chen[1], Balbir Pabla[1], Wanmin Gong[1], Craig Stroud[1], Christopher Sioris[1], Kerry Anderson[2], Philip Cheung[1], Junhua Zhang[1], Jason Milbrandt[3]

[1]Air Quality Research Division, Atmospheric Science and Technology Directorate, Environment and Climate Change Canada, 4905 Dufferin Street, Toronto, M3H 5T4, Canada

[2]Natural Resources Canada (emiritus), Summerland, British Columbia, Canada.

[3]Meteorological Research Division, Atmospheric Science and Technology Directorate, Environment and Climate Change Canada, 2121 Trans Canada Highway, Montreal, Canada

*Correspondence to*: Paul A. Makar (paul.makar@canada.ca)

**Abstract.** The influence of both anthropogenic and forest fire emissions, and their and subsequent chemical and physical processing, on the accuracy of weather and air-quality forecasts, was studied using a high resolution, fully coupled air-quality model. Simulations were carried out for the period 4 July through 5 August 2019, at 2.5-km horizontal grid cell size, over a 2250 x 3425 $km^2$ domain covering western Canada and USA, prior to the use of the forecast system as part of the FIREX-AQ ensemble forecast. Several large forest fires took place in the Canadian portion of the domain during the study period. A feature of the implementation was the incorporation of a new on-line version of the Canadian Forest Fire Emissions Prediction System (CFFEPSv4.0). This inclusion of thermodynamic forest fire plume-rise calculations directly into the on-line air-quality model allowed us to simulate the interactions between forest fire plume development and weather.

Incorporating feedbacks resulted in improvements in most metrics of both air-quality and meteorological model forecast performance, through comparison of no-feedback and feedback simulations with surface, radiosonde, and satellite observations. For the meteorological simulations, these improvements occurred at greater than the 90% confidence level. Relative to the climatological cloud condensation nuclei and aerosol optical properties used in the no-feedback simulations, the fully coupled model's aerosol indirect and direct effects were shown to result in feedback loops characterized by increased surface temperatures, decreased lower troposphere temperatures, and increased lower troposphere cloud droplet and raindrop number densities. The aerosol direct and indirect effect reduced oceanic cloud droplet number densities and increased oceanic rain drop number densities, relative to the no-feedback climatological simulation. The aerosol direct and indirect effects were responsible for changes to the aerosol concentrations at greater than the 90% confidence level throughout the model domain, and to $NO_2$ and $O_3$ concentrations within forest fire plumes.

The simulations show that incorporating aerosol direct and indirect effect feedbacks can significantly improve the accuracy of weather and air quality forecasts, and that forest fire plume rise calculations within a fully coupled model changes the predicted fire plume dispersion and emissions, the latter through changing the meteorology driving fire behaviour and growth.





## 1 Introduction

Atmospheric aerosol particles may be emitted (primary particles) or result from the condensation of the products of gas-phase oxidation reactions (secondary aerosol). With increasing transport time from emission sources, the processes of coagulation (colliding particles stick adhere creating larger particles) and condensation (low volatility gases condense to particle surfaces) tend to result in particles which have a greater degree of internal mixing (internal homogeneous mixtures). Primary and near-source particles are more likely to have a single or a smaller number of chemical constituents (external mixtures).

Atmospheric particles also modify weather through well-established pathways. Under clear sky conditions, the particles may absorb and/or scatter incoming light, depending on their size, shape, mixing state (internal, external or combinations) and their composition. The presence of the particles themselves may thus affect the radiative budget of the atmosphere, resulting in either positive or negative climate forcing (i.e. the absorption of a greater amount of incoming solar radiation versus increased scattering reflection of that radiation back out into space, a process known as the Aerosol Direct Effect; ADE). Aerosols can also alter the atmospheric radiative balance through interactions with clouds, this influence being referred to as the Aerosol Indirect Effect (AIE). Three broad classes of categories by which cloud/aerosol interactions take place (Oreopoulos *et al.*, 2020) include the first indirect effect, where higher aerosol loadings resulting in increasing numbers of cloud droplets with smaller sizes, hence increasing cloud albedo (Twomey *et al.*, 1977), the second indirect effect, where higher aerosol loadings suppress the collision-coalescence activity of the smaller droplets, reducing precipitation/drizzle, changing cloud heights, and changing cloud lifetime in warm clouds (Albrecht, 1989), and aerosol "invigoration" of storm clouds, where higher aerosol loadings may result in delayed glaciation of cloud droplets, in turn leading to greater latent heat release and stronger convection (Rosenfeld *et al.*, 2018).

The uncertainties associated with the ADE and particularly AIE account for a large portion of the uncertainties in current climate model predictions for radiative forcing between 1750 and 2011 (Mhyre *et al.*, 2013). Carbon dioxide is believed to have a positive (warming) global radiative forcing of approximately $1.88 +/- 0.20$ $Wm^{-2}$, while the direct and indirect effects both have nominal values of approximately $-0.45$ $Wm^{-2}$, with uncertainty ranges encompassing $-0.94$ to $+0.07$ and $-1.22$ to $0.0$ $Wm^{-2}$ respectively. These uncertainties have spurred research designed to better characterize the ADE and AIE, and reduce these uncertainties, through both observations and atmospheric modelling. Observational studies of the ADE have established its large impact; for example, high aerosol loading over Eurasian boreal forests has been found to double the diffuse fraction of global radiation (i.e. increased scattering), a change sufficient to affect plant growth characterized via gross primary production (Ezhova *et al.*, 2018). Aerosol assimilation of Geostationary Ocean Color Imager Aerosol Optical Depth (AOD) observations into a coupled meteorology-chemistry model showed that South Korean AOD values increased by as much as 0.15 with the use of assimilation; these increases corresponded to a local $-31.39$ $W$ $m^{-2}$ reduction in solar radiation received at the surface, and reductions in planetary boundary layer height, air temperature, and surface wind speed over land, and a deceleration of vertical transport (Jung *et al.*, 2019). Other studies in East Asia have shown ADE decreasing local shortwave reaching the surface by $-20$ $Wm^{-2}$ (Wang *et al.*, 2016), as well as significant changes in surface particulate matter and gas concentrations in response to these radiation changes.



However, one commonality amongst the recent studies of the ADE for air-quality models is a tendency towards
negative biases in predicted aerosol optical depths, potentially indicating systematic under-predictions in aerosol mass,
aerosol size, and/or inaccuracies in the assumptions for shape and/or mixing state.  Mallet *et al. (*2017) noted this
negative bias for regional climate model AOD predictions associated with large California forest fires compared to
OMI and MRIS satellite observations.  Palacios-Pena *et al. (*2018) noted that high AOD events associated with forest
fires were under-predicted by most models in a study employing a multi-regional-model ensemble.  The chosen AOD
calculation methodology and mixing state assumptions employed in models also plays a role in systematic biases:
Curci *et al. (*2015) compared aerosol optical depths, single scattering albedos, and asymmetry factors at different
locations to observations, varying the source model for the aerosol composition, as well as the mixing state
assumptions used in generating aerosol optical properties, for Europe and North America.  AODs were biased low by
a factor of two or more, regardless of model aerosol inputs or mixing state assumptions at 440 nm, single scattering
albedos were biased low by up to a factor of two, with the poorest performance for "core-shell" approaches, while
asymmetry factor estimates showed no consistent bias relative to observations.  However, the assumed mixing state
was clearly a controlling factor in the negative biases; the AOD predictions closest to the observations at 440 nm
assumed an external mixture with particle sulphate and nitrate assumed to grow hygroscopically as pure sulphuric
acid, lowering their refractive index with increasing aerosol size.  This mixing state assumption and the different
homogeneous mixture assumptions gave the best fit for single scattering albedo relative to observations.   While not
commenting on aerosol direct effect implications, Takeishi *et al. (*2020) noted that forest fire aerosols increase particle
number concentrations but reduce their water uptake (hygroscopicity) relative to anthropogenic aerosols, with the
latter effect reducing the resulting cloud droplet numbers by up to 37%.  Mixing state and hygroscopicity properties
of aerosols were thus shown to have a controlling influence on the ADE.
The AIE has often been shown to be locally more important for the radiative balance than ADE in terms of magnitude
of the radiative forcing and response of predicted weather to AIE and ADE (Makar *et al.,* 2015(a); Jiang *et al.,* 2015;
Nazarenko *et al.,* 2017).  Several recent studies have attempted to characterize the relative importance of the AIE with
the use of multi-year satellite observations, sometimes making use of models and data assimilation.  Saponaro *et al.*
*(*2017) used MODIS/Aqua linked observations of aerosol optical depth and Ångström exponent to various cloud
properties, noting that the cloud fraction, cloud optical thickness, liquid water path, and cloud top height all increased
with increasing aerosol loading, while cloud droplet effective radius decreased, with the effects dominating at low
levels (between 900 to 700 hPa).  Zhao *et al.* (2018) examined 30 years of cloud and aerosol data (1981-2011), and
found that increasing aerosol loading up to   AOD < 0.08 increased cloud cover fraction and cloud top height, while
further increases in aerosol loading (AOD  from 0.08 to 0.13) resulted in higher cloud tops, and larger cloud droplets.
In polluted environments (AOD > 0.30) cloud droplet effective radius, optical depth and water path; cloud droplet
effective radius increased with increasing AOD.  The first ADE was most sensitive to AOD in the AOD range 0.13 to
0.30; and the reduction of precipitation efficiency associated with the second aerosol indirect effect occurred for AODs
between 0.08 and 0.4, in oceanic areas downwind of continental sources.
However, sources of uncertainty in AIE estimates persist, in part due to the number of poorly understood processes
contributing to the atmospheric response to the presence of aerosols.  Nazerenko *et al. (*2017) showed that short-term



atmospheric radiative changes were reduced in magnitude when sea-surface temperature and sea-ice coupling was
included in climate change simulations. Suzuki *et al. (*2019) showed that the vertical structure of atmospheric aerosols,
as well as their composition, had a significant influence on radiative forcing. Penner *et al. (*2018) and Zhu *et al. (*2020)
examined the impact of aerosol composition on cirrus clouds via ice nucleation, finding negative forcings for most
forms of soot, but a contrary impact of secondary organic aerosols. Rothenburg *et al. (*2018) noted that tests of aerosol
activation schemes carried out under current climate conditions had little variability, but had much greater variability
for pre-industrial simulations, implying that the available data for evaluation using current conditions may poorly
constrain ADE and AIE parameterizations used in simulating in past climates.
Forest fires are of key interest for improving the understanding and representation of ADE and AIE in models, due to
the large amount of aerosols released during these biomass burning events. Forest fire emissions and interactions with
weather are also of interest due to the expectation that the meteorological conditions resulting in forest fires may
become more prevalent in the future under climate change (Hoegh-Guldberg *et al.*, 2018). Observations of aerosol
optical properties during long-range transport events of North American forest fire plumes to Europe showed 500 nm
AOD values of 0.7 to 1.2 over Norway, with Ångström exponents exceeding 1.4 and absorbing angstrom exponents
ranging from 1.0 to 1.25, along with single scattering albedos greater than 0.9 at the surface and up to 0.99 in the
column over these sites (Markowicz *et al.,* 2016). Biomass burning was shown to have a specific set of optical
properties relatively independent of fuel type for three different types of biomass burning in China (cropland), Siberia
(mixed forest) and California (needleleaf forest). The increase in upward radiative forcing at the top of the atmosphere
due to fires being linearly correlated to AOD (R from 0.48 to 0.68), with slopes covering a relatively small range from
20 to 23 W m$^{-2}$ unit AOD$^{-1}$. O'Neill *et al. (*2001) showed that forest fires have a profound impact on aerosol optical
depth in western Canada, accounting for 80% of the summer AOD variability in that region, with a factor of three
increase in AOD levels from clear-sky to forest fire plume conditions. O'Neill *et al. (*2001)'s analysis of TOMS
AVHRR and GOES imagery suggested that forest fire aerosols increase in size with increasing downwind distance,
due to secondary aerosol aging and condensation chemistry. We note here that reanalyzing the data presented in
O'Neill *et al.* (2001) results in a linear relationship between fine mode particle effective radius ($r_{eff}$, μm) and the base
10 logarithm of distance from the fires (D, km) of $r_{eff} = 0.0106 \, log_{10}(D) + 0.1163, R^2 = 0.18$). Mallet *et al.*
*(*2017) simulated AODs in the range 1 to 2 for biomass burning events, and also noted changes in direct radiative
forcing at the top of the atmosphere from positive to negative in both model results and simulations, with increasing
downwind distance from the sources. Lu *et al. (*2017) carried out simulations with 5-km horizontal grid spacings for
the eastern Russia forest fires of 2002 assuming an internal mixture for emitted aerosols with the WRF-CHEM model,
and noted impacts on cloud formation for two different periods. The first period was characterized by high cloud
droplet and small ice nuclei numbers, where the fire plumes reduced cloud rain and snow water content, large scale
frontal system dynamics were altered by smoke, and precipitation was delayed by a day. The second period was
characterized by high numbers for cloud droplets and ice nuclei, where the fire plumes reduced rain water content,
increased snow water content, and precipitation locations changed locally across the simulation domain. Russian
forest fire simulations for 2010 with suites of fully coupled air-quality models (Makar *et al.,* 2015; Palacios-Pena *et*



*al.,* 2018; Baro *et al.,* 2017) showed substantial local impacts, such as reductions in average downward shortwave
radiation of up to 80Wm$^{-2}$ and temperature of -0.8 ºC (Makar *et al.,* 2015(a)).
Given the above developments in direct and indirect parameterizations, and the increasing amount of information
available for estimating forest fire emissions, the impact of forest fires on weather, in the context of weather
forecasting, is worthy of consideration. Air-quality model predictions of forest fire plumes have been provided to the
public under operational forecast conditions of time- and memory-space limited computer resources (e.g. Chen *et al.,*
2019; James *et al.,* 2018; Ahmadov *et al.,* 2019, Pan  *et al.,* 2017). These simulations make use of satellite retrievals
of forest fire hot-spots, climatological data on the extent of area burned by land use type, databases of fuel type linked
to emission factors, and an *a priori* weather forecast to provide the meteorological inputs required to predict forest
fire plume rise. The latter point is worthy of note in the context of the direct and indirect feedback studies noted above
– both climate and weather simulations with prescribed forest fire emissions have consistently resulted in large
perturbations of weather patterns in the vicinity of the forest fires. However, the approaches for predicting forest fire
plume rise up until now have relied on weather forecast information lacking those meteorological feedback effects.
The connection of the ADE and AIE within an air-quality and weather forecast model is referred to as "coupling",
with such a model being "fully-coupled". However, we note that in the field of very high resolution forest fire
behaviour modelling, coupling of biomass burning with the atmosphere has also been defined as the interaction of
dynamic meteorology with the heat released by the fire, where the initial meteorology determines fire spread on the
landscape. This in turn, modifies the temperature and wind fields, in turn affecting future fire spread (Clark *et al.,*
1996, Linn *et al.,* 2002). The coupling presented in the present paper refers to that between the aerosols released by
fires and other sources to meteorology through the ADE and AIE, with the resulting changes in meteorology in turn
influencing fire behaviour (fire intensity, fuel consumption, etc.), in turn influencing emissions height and distribution,
closing this feedback loop.
A key consideration in parameterizing the AIE (via aerosol-cloud interaction) is the manner in which the cloud
condensation process is represented in the meteorological component of the modelling system. In numerical weather
prediction (NWP) models, clouds and precipitation are represented by a combination of physical parameterizations
that are each targeted at a specific subset of moist processes. These include "implicit" (subgrid-scale) clouds generated
by the boundary layer and the convection parameterization schemes (e.g Sundqvist, 1988), and "explicit" clouds from
the grid-scale condensation scheme (Milbrandt and Yau, 2005(a,b), Morrison and Milbrandt, 2015, Milbrandt and
Morrison, 2016). Depending on the model grid these "moist physics" schemes vary in their relative importance.
However, regardless of the horizontal grid cell size, the grid-scale condensation scheme plays a crucial role in
atmospheric models, though to different degrees and using different methods, depending on the grid spacing and the
corresponding relative contributions of the implicit schemes. A grid-scale condensation scheme will in general consist
of the following three components: 1) a subgrid cloud fraction parameterization (CF, or cloud "macrophysics"
scheme); 2) a microphysics scheme; and 3) a precipitation scheme (Jouan *et al.*, 2020). The cloud fraction (CF) is the
percentage of the grid element that is covered by cloud (and is saturated), even though the grid-scale relative humidity
may be less than 100%. The microphysics parameterization computes the bulk effects of a complex set of cloud
microphysical processes. If precipitating hydrometeors are advected by the model dynamics, the precipitation is said



to be *prognostic*; if precipitation is assumed to fall instantly to the surface upon production, it is considered *diagnostic*.
The precipitation "scheme" is not a separate component per se, since it simply reflects the level of detail in the
microphysics parameterization, but it is a useful concept to facilitate the comparison of different grid-scale
condensation parameterizations.
With a wide range of grid cell sizes in current NWP models, there is a wide variety of types of condensation schemes
and degrees of complexity in their various components. For example, cloud-resolving models (with grid spacing on
the order of 1 km or less) have typically used detailed bulk microphysics schemes (BMSs), with prognostic
precipitation, and no diagnostic or prognostic CF component (i.e. the CF is either 0 or 1). Large-scale global models
use condensation parameterizations, sometimes referred to as "stratiform" cloud schemes, typically with much simpler
microphysics and diagnostic precipitation, but with more emphasis on the details of the CF. However, with continually
increasing computer resources and decreasing grid spacing (both in research and operational prediction systems), the
distinction between schemes designed for specific ranges of model resolutions is disappearing and condensation
schemes are being designed or modified to be more versatile and usable across a wider range of model resolutions
(e.g. Milbrandt and Morrison, 2016).
Aerosol-cloud interactions and feedback mechanisms are difficult to represent in grid-scale condensation schemes
with very simple microphysics components. For example, to benefit from the predicted number concentrations of
cloud condensation nuclei and ice nuclei, the microphysics needs to be double-moment (predicting both mass and
number) for at least cloud droplets and ice crystals, respectively. Until recently, detailed BMSs were only used at
cloud resolving scales, hence requiring these relatively high resolutions to be recommended in feedback modelling.
In recent years, multi-moment BMSs have been used in operational NWP for model grid spacings of 2-4 km (e.g.
Seity *et al*., 2010, Pinto *et al*., 2015, Milbrandt *et al*., 2016). Further, condensation schemes with detailed microphysics
are starting to use non-binary CF components (e.g. Chosson *et al.*, 2014, Jouan *et al.,* 2020), thereby allowing detailed
microphysics to be used at larger scales, and hence allowing the same indirect feedback parameterizations to be used
at multiple scales. Nevertheless, the expectation is that detailed parameterization will provide a more accurate
representation of cloud formation at the near cloud-resolving scales, without the complicating aspect of a diagnostic
CF, motivating the use of km-scale grid spacing for feedback studies.
The formation of secondary aerosols from complex chemical reactions are another key consideration in feedback
forecast implementation, given the impact of aerosol composition on aerosol optical and cloud formation properties,
as described above.
In the sections which follow, we describe our high resolution, fully coupled air-quality model with on-line forest fire
plume rise calculations, which was created as part of the FIREX-AQ air-quality forecast ensemble
(https://www.esrl.noaa.gov/csl/projects/firex-aq/), to address the following questions:
(1) Will a fully coupled model of this nature provide improved forecasts of *both* weather and air-quality, using

standard operational forecast evaluation tools, techniques and metrics of forecast confidence? That is, despite the

uncertainties in the literature as described above, are these processes sufficiently well described in our model that

their use results in a formal improvement in forecast accuracy?



(2) Are the changes in forest fire plume rise associated with implementing this process directly within a fully coupled model sufficient to result in significant perturbations to weather predictions and to chemistry? What are these perturbations?

We employ our fully coupled model with 2.5-km grid cell size domain covering most of western North America, and compare model results to surface meteorological and chemical observations, and to vertical column observations of temperature and aerosol optical depth (AOD), in order to quantitatively evaluate the effect of feedback coupling of the ADE and AIE on model performance. We then compare feedback and no-feedback simulations to show the impacts of the ADE and AIE feedbacks on cloud and other meteorological predictions, and on key air quality variables (particulate matter, nitrogen dioxide, and ozone). We begin our analysis with a description of our modelling platform.

## 2 Model Description

### 2.1 GEM-MACH

The Global Environmental Multiscale – Modelling Air-quality and CHemistry (GEM-MACH) model in its fully coupled configuration has been described elsewhere (Makar *et al.,* 2015a,b; Gong *et al.,* 2015, 2016). Briefly, the model combines the Environment and Climate Change Canada Global Environmental Multiscale weather numerical weather prediction model (GEM, Cote *et al.,*, 1998, Girard *et al.,* 2014) with gas and particle process representation using the on-line paradigm, with options for climatological versus full coupling between meteorology and chemistry. In GEM-MACH's climatological coupling configuration, prescribed, invariant climatological values for aerosol optical properties and cloud condensation nuclei (CCN) are employed within the model's radiative transfer and cloud microphysics modules. In the full coupling configuration, the ADE is simulated using GEM-MACH's predicted aerosol loading and Mie scattering using a binary water-dry aerosol homogeneous mixture assumption, at the 4 wavelengths employed by GEM's radiative transfer algorithms, and at additional wavelengths for diagnostic purposes. The full coupling also includes the AIE by simulating aerosol-cloud interaction via explicit droplet nucleation using the algorithm of Abdul-Razzak and Ghan (2002) along with on-line aerosol composition and size (Gong *et al.,* 2015). This droplet nucleation replaces the decoupled model's existing droplet nucleation calculation in the Predicted Particle Properties (P3) microphysics scheme (Morrison and Milbrandt, 2015, Milbrandt and Morrison, 2016). The latter assumes an invariant aerosol population of a single lognormal size distribution (with a geometric mean diameter of 100 nm and total aerosol number concentration of 300 cm$^{-3}$, consisting of pure ammonium sulphate; Morrison and Grabowski, 2008). The prognostic cloud droplet number and mass mixing ratios from the P3 microphysics are then transferred back to the chemistry module for using in cloud processing of gases and aerosols (cloud scavenging and chemistry) calculations, completing the AIE feedback process loop (Gong *et al.,* 2015).

The chemistry modules of GEM-MACH also include process representation for gas-phase chemistry (ADOMII mechanism, 42 gas species, Stockwell *et al.,* 1989), cloud processing including aqueous chemistry, scavenging of gases and aerosols, below-cloud removal and wet deposition (Gong *et al.,* 2015), particle microphysics employing a sectional size distribution and 8 chemical species (Gong *et al.,* 2003), particle inorganic thermodynamics (Makar *et al.,* 2003), the formation of secondary organic aerosols using a modified yield approach (Stroud *et al.,* 2018), process



representation for surface fluxes as a boundary condition on the solution for vertical diffusion, and semi-Lagrangian
advection for transported chemical tracers.
The specific base model version employed in these simulations is GEMv4.9.8/GEM-MACHv2 (Moran *et al.,* 2018),
incorporating the following additional improvements in addition to those noted above: (a) the AIE parameterization
was modified for use with the P3 cloud microphysics scheme; (b) forest canopy shading and turbulence was
parameterized following Makar *et al.,* (2017); (c) anthropogenic plume rise was parameterized through calculating
residual buoyancy of the rising plume (Akingunola *et al.,* 2018); (d) emissions of crustal material undergo a
meteorological modulation with crustal material emissions being inhibited when the soil water content is predicted to
be greater than 10%; (e) emissions/deposition of $NH_3$ are implemented using a bidirectional flux parameterization
(Whaley *et al.,* 2018; Zhang *et al.,* 2003); (f) $CH_4$ is treated as a reactive and emitted tracer; (f) the KPP-generated
RODAS-3 solver (Sandu and Sander, 2006) is used for the solution of ADOMII gas-phase mechanism (Stockwell
and Lurmann, 1989); (g) MODIS retrievals were used to create monthly leaf area index values for use in the model's
biogenic emissions, forest canopy shading and turbulence, and deposition algorithms; (h) a parameterization for the
impacts of vehicle-induced turbulence on vertical diffusive transport was employed (Makar *et al.,* 2020).
Simulations were carried out with a 2.5-km horizontal grid cell spacing over a 900 x 1370 grid cell domain, covering
most of western Canada and the USA (Figure 1). The meteorological boundary conditions for the simulation were a
combination of 10-km GEM forecasts updated hourly (themselves originating in data assimilation analyses of real-
time weather information), and 2.5-km GEM simulations employing, in the northern portion of the 2.5-km domain,
the Canadian Land Data Assimilation System (Carrera *et al.,* 2015), to better simulate surface conditions. Both
"feedback" and "no feedback" simulations were carried out on a 30-hour forecast cycle (Figure 2). Following the
usual practice for weather forecasts, the analysis-driven meteorological forecasts at 10 km resolution were updated
operationally every 24 hours at 12 UT. These 10 km resolution forecasts were followed by a 30-hour meteorology-
only forecast at 2.5-km resolution on the high resolution domain of Figure 1, the last 24 hours of which were used as
meteorological initial and boundary conditions for the 24-hour 2.5-km fully-coupled GEM-MACH simulation. These
two stages of meteorology-only simulations were carried out to prevent chaotic drift from the observed meteorology,
and to allow spin-up time for the cloud fields of that meteorology to reach equilibrium (6-hour timeframe). Chemical
lateral boundary conditions were taken from climatologies based on ECMWF MACC-II global atmospheric chemical
composition modelling and reanalysis (Inness *et al.,* 2013). Chemical initial concentrations for each consecutive
forecast within the 2.5- km GEM-MACH model domain were "rolled over" or "daisy-chained" between subsequent
forecasts without chemical data assimilation. Forecast performance scores presented here are for the inner 2.5-km
domain from this set of linked 24 forecast simulations, mimicking operational forecast conditions.
**2.2 CFFEPS Version 4.0: On-line forest-fire plume rise calculations**
In addition to the above algorithm improvements relative to GEM-MACH implementations, this model system setup
has incorporated the first on-line calculation of forest-fire plume-rise by energy balance driven using on-line
meteorology, in a new version of the Canadian Forest Fire Emissions Prediction System (CFFEPS). The algorithms





289 of CFFEPSv2.03 are described in detail and evaluated elsewhere (Chen *et al.,* 2019), but will be outlined briefly here,

290 as well as subsequent modifications to this forest fire emissions processing module.

291 CFFEPS combines near-real-time satellite detection of forest fire hotspots with national statistics of burn areas by

292 Canadian province and by specific fuel type across North America. CFFEPS assumes persistence fire growth in the

293 subsequent 24- to 72-hour forecasts with hourly fuel consumed calculated (kg m$^{-2}$), based on GEM forecast

294 meteorology and predicted fire behaviour in grid cells representing fire locations. The modelled fire fuel consumption

295 is then linked with combustion-phase specific emission factors (g kg$^{-1}$) for fire specific emissions and chemical

296 speciation. Fire energy associated with the modelled combustion process is also estimated, and is used in conjunction

297 with *a priori* forecasts of meteorology within the column to determine plume rise. In its off-line/non-coupled

298 configuration (Chen *et al*., 2019), CFFEPS carries out residual buoyancy calculations at five preset pressure levels

299 (surface, 850, 700, 500, 250 mb). CFFEPS predicts plume injection heights, which are in turn used to redistribute the

300 mass emissions below the plume top to the model hybrid levels. This approach employed in CFFEPSv2.03 provided

301 a substantial improvement in forecast accuracy relative to the previous approach employing modified Briggs (Briggs,

302 1965, Pavlovic *et al.,* 2016) plume rise formulae in the offline GEM-MACH forecast system (Chen *et al.,* 2019).

303 However, other work has shown the substantial impact of large forest fires on regional weather (Makar *et al.,* 2015a;

304 Palacios-Pena *et al.,* 2018, Baro *et al.,* 2017), including changes to the surface radiative balance and atmospheric

305 stability. These findings imply that plume rise calculations employing an *a priori* weather forecast lacking the impact

306 of fire plumes via the ADE and AIE may not accurately predict the weather conditions critical to subsequent forest

307 fire plume rise prediction. In order to study this possibility, and to allow forest fire plumes to influence weather and

308 hence subsequent fire spread/growth, several changes were made to CFFEPS implementation, resulting in version 4.0

309 of CFFEPS, used here. The process flow within CFFEPSv2.03 versus CFFEPSv4.0 are compared in Figure 3. The

310 original C language CFFEPSv2.03 code was converted to FORTRAN90, and following successful off-line

311 comparisons to the original code, was then integrated as an on-line subroutine package within GEM-MACH itself,

312 with the near-real-time satellite hotspot data and location fuel parameters being read into GEM-MACH directly

313 (CFFEPSv4.0 is this new on-line package). A key advantage of the CFFEPSv4.0 subroutine integration within GEM-

314 MACH is that the residual buoyancy calculations for plume injection heights are now carried out over the model

315 hybrid model layers, rather than the five coarse resolution, prescribed pressure levels of CFFEPSv2.03, making

316 complete use of GEM-MACH's detailed vertical structure. Additionally, CFFEPSv4.0 allows plume rise calculations

317 to be updated during model runtime. When GEM-MACH is run in fully-coupled mode, the ADE and AIE

318 implementations allow model-generated aerosols to modify the predicted meteorology, in turn influencing predicted

319 fire emissions and plume rise, closing these feedback loops. The on-line implementation of CFFEPSv4.0 thus allows

320 us to investigate the effects of meteorology on subsequent forest fire plume development, the changes to modelled

321 aerosol compositions, and, ultimately, the feedbacks to weather.

322 The formation of particles from forest fires affects meteorology on the larger scale via the ADE and AIE, in turn

323 modifying the regional scale atmospheric features affecting fire growth, such as the temperature profiles below forest

324 fire plumes. However, we note that the local scale weather modifications due to the addition of forest fire heat to the

325 atmosphere are not yet incorporated into fire spread or GEM microphysics. Specifically, when the feedback version





of GEM-MACH incorporating CFFEPSv4.0 is used in its fully coupled configuration, CFFEPSv4.0 calculates forest
fire plume rise using the meteorological predictions which include the ADE and AIE generated by forest fire aerosols
on atmospheric stability from the current fully-coupled model timestep. The resulting added aerosol mass due to the
fire in turn affects the meteorology through ADE and AIE, closing this feedback loop. To the best of our knowledge,
this is the first implementation of a dynamic forest fire plume injection height scheme incorporated into a fully coupled
high-resolution, air quality forecast modelling system. The impact of this feedback on both weather and air-quality
can be substantial, as we show in the following sections.
The locations of the daily forest hotspots detected during the study period, and the corresponding magnitude of the
daily PM2.5 emissions generated by CFFEPS for each hotspot are shown in Figure 4. Individual hotspots with the
highest magnitude emissions are located in the state of Nevada (Figure 4(a), southern boxed region). However, the
largest ensemble emissions from a suite of hotspots occurs in northern Alberta (Figure 4(a), northern boxed region).
Expanded views of the northern Alberta and Nevada hotspots are shown in Figure 4(b,c) respectively – the use of
smaller symbols shows that the Alberta hotspots are groups representing large spreading fires, which overplotted in
Figure 4(a), while the Nevada hotspots indicate single fires of small spatial extent and duration rather than larger
spreading fires. The Alberta fires are thus the most significant sources of forest fire emissions in the study domain for
the period analyzed here.

**2.2 Feedback and No-Feedback Simulations**

Two simulations were carried out for the period July 4$^{th}$ through August 5$^{th}$ 2019; a "feedback" (ADE and AIE
feedbacks enabled – fully coupled model) and a "no-feedback" simulation (ADE and AIE make use of GEM's
climatological aerosol radiative and CCN properties – the decoupled model). During this period, five large forest fires
took place in the northern portion of the modelling domain. The two parallel combined meteorology and air-quality
forecasts in the fully coupled model with/without ADE and AIE coupling were evaluated using the US EPA AIRNOW
data (https://www.airnow.gov) and Environment and Climate Change Canada's EMET and ARCAD operational
forecast evaluation systems, respectively. Following evaluation, the simulation mean values of hourly meteorological
and chemical tracer predictions were compared to analyze the impact of fully coupled ADE and AIE feedbacks on
both sets of fields.

**3 Model Evaluation**

**3.1 Meteorology Evaluation**

Surface meteorological conditions were evaluated at three-hour intervals from the start of both of the two sets of paired
24-hour forecasts using standard metrics of weather forecast performance including mean bias (MB), mean absolute
error (MAE), root mean square error (RMSE), correlation coefficient (R) and standard deviation ($\sigma$). In all
comparisons, a 90% percent confidence level assuming a normal distribution was used to identify statistically different
results between forecast simulations. Note that 90% confidence levels are commonly used in meteorological forecast





evaluation, with values of 80% to 85% recommended (Pinson and Kariniotakis, 2004) and up to 90% used (Luig *et*
*al.,* 2001) for variables such as wind speed, rather than the 95% or 99% confidence levels in other fields, in recognition
of the difficulties inherent in prognostic forecasts of the chaotic weather system.    Here, the confidence range
formulation of Geer (2014) has been applied using a 90% confidence level in model predictions, with the statistical
measures considered different at the 90% confidence level when the 90% confidence ranges do not overlap.    The
surface meteorological evaluations shown here only include those variables and metrics where results were
significantly different at the 90% confidence level.
Several model forecast output variables were evaluated and the surface variables showing statistically significant
differences relative to observations at the 90% confidence level included: 2 m temperature, surface pressure, 2 m
dewpoint temperature, 10 m wind speed, sea-level pressure, and accumulated precipitation (the latter in 3 different
metrics).  The comparisons are shown as time series in three-hourly intervals as a function of forecast hour prediction
time forward from forecast hour 0, for grid cells corresponding to measurement locations in Figures 5, 6, 7, 8, 9, 10,
and 11 for each of these quantities, respectively.  Note that these statistics measure domain-wide performance, across
all of the reporting stations within the model domain, during the sequence of 24-hour forecasts comprising the
simulation period.  The duration of the time series in these comparison figures is thus a function of the duration of the
contributing forecasts.
Figure 5 shows an example analysis for surface temperature bias for the entire model domain.  Figure 5(a) shows the
average model MB time series across all stations and all forecasts at the given forecast hours, while Figure 5(b) shows
the corresponding difference in the MB absolute values.  The difference plot in Figure 5(b) shows the feedback – no-
feedback scores, such that scores below the zero line indicate superior performance of the feedback forecast, while
those above the zero line indicate superior performance of the no-feedback forecast.  Here, the feedback forecast was
statistically superior at forecast hours 3, 6, 15 and 18 at the 90% confidence level at these forecast hours, and both
simulations were at par (differences below the 90% confidence level) at hours 9, 12, 21 and 24.  The feedback forecast
thus has superior performance, at greater than 90% confidence, over half of the forecast hours evaluated within the
domain, during the simulation period.
All of the metrics for which surface temperature forecast performance differed at the 90% confidence level are shown
in Figure 6.  In addition to MB, the scores for MAE, and RMSE showed superior forecast performance for the feedback
relative to the no-feedback case at the 90% confidence level for hours 6 through 15, while the improvement for the
correlation coefficient was only reached the 90% confidence level at hours 6 and 12.
Model 10-m windspeed forecasts were also improved with the incorporation of feedbacks for hour ranges between
hour 3 and hour 12, depending on the metric, with the longest duration improvement for MB, MAE, and RMSE, and
the shortest duration for correlation coefficient and standard deviation (Figure 9).   A marginal performance
degradation of the feedback forecast at close to 90% confidence at hours 15-18 can also be seen for root mean square
error, correlation coefficient, and standard deviation in this Figure.
Precipitation forecast performance from the two simulations varied depending on the metric chosen (Figure 11). The
metrics in this case are based on the number of coincident precipitation "events" versus "non-events" as shown in
contingency Table 1.



The Heidke skill score { $HSS = 2\,(AD - BC)/[(A + C)(C + D) + (A + B)](B + D)$ } measures the fractional
improvement of the forecast over the number correct by chance. The Frequency Bias { $FB = (A + B)/(A + C)$ }
measures the frequency of event over-forecasts (FB>1) versus event under-forecasts (FB<1). The Equitable Threat
score { $ETS = (A - \tilde{A})/(A + C + B - \tilde{A})$, where $\tilde{A} = (A + B)(A + C)/(A + B + C + D)$} measures the observed
and/or forecast events that were correctly predicted. Following standard practice at Environment and Climate Change
Canada, the HSS is used as a measure of total precipitation accumulated over a 6-hour interval, with no lower limit
on the amount of precipitation defining an "event", while FB and ETS define precipitation "events" as being those
with greater than 2mm / 6 hours – consequently FB and ETS have a smaller number of data points for comparison
than HSS.
Figure 11 shows improvements to the fully coupled precipitation forecast at the 90% confidence level were seen for
the HSS 6-hour accumulated metric by hours 18 and 24, while the frequency bias index of 6-hour accumulated
precipitation showed degradation at hours 6 and 24, and the equitable threat score of 6-hour accumulated precipitation
showed degradation at hour 24. As is noted above, the latter two metrics employed a minimum 6-hour precipitation
threshold of 2 mm prior to comparisons (this is the reason for the reduced number of points available for comparison
in Figure 11(b,c) relative to Figure 11(a)). These findings suggest that the fully coupled model's trend towards
improved total precipitation over time (Figure 11(a)) is the result of improved performance for relatively low-level
precipitation events (< 2mm 6hr$^{-1}$), offsetting a degradation of performance for higher level precipitation events.
Precipitation events have thus become more frequent, but "lighter" with the use of the feedback parameterizations.
The meteorological forecast performance metrics with statistically significant differences for surface pressure,
dewpoint temperature, and sea-level pressure are shown in Figures 7, 8, and 10 respectively. The model performance
differences in these three Figures show a similar pattern: a degradation in performance with the use of feedbacks at
hour 3, with the differences between the two forecasts either dropping below the 90% confidence level, or the feedback
forecast showing an improvement by hour 6, followed by several hours in which the feedback forecast has a superior
performance. The duration of this latter period varies between the metrics, from up to 18 hours for MAE for surface
pressure (Figure 7(b)) to 6  hours for the standard deviation of dew-point temperature (Figure 8(d)).
We believe that the initial loss of performance for the feedback forecast may represent a form of "model spin-up" that
may be unique to fully coupled models, but may be affected or improved with further adjustments to the forecast
cycling setup for the chemical species. As noted earlier (Figure 2), in order to prevent chaotic drift from observed
meteorology, we made use of a 30-hour 2.5-km resolution analysis-driven weather forecast to update our fully coupled
model's initial meteorology at hour zero of each 24 hour forecast. The cloud fields provided as initial conditions at
hour zero include observation analysis for the 6 hours prior to hour zero - these have reached a quasi-equilibrium in
the high-resolution weather forecast (Figures 2(b,e)) by the time they are used as initial and boundary conditions in
the fully coupled model (Figure 2(c,f)). However, the fully coupled model's *aerosol* fields at hour zero, used to
initialize the subsequent forecast (Figure 2, dashed blue arrow), still reflect the locations of aerosol-cloud interactions
in the previous fully coupled simulation. We believe that the initial three to six hours of feedback forecast degradation
represents the time required for the fully coupled model to reach a new equilibrium consistent between both its aerosol
and the cloud fields.





One possible solution for this model spin-up inconsistency would be to eliminate the intermediate driving 2.5-km
meteorological simulation in favour of a longer 30-hour fully coupled forecast with the first six hours removed as
spin-up (i.e. extend the duration of steps (c) and (f) in Figure 2 to 30 hours, starting at UT hour 6). The duration of
the forecast experiments carried out here was limited to 24 hours due to limited computational resources, and, more
importantly, the operational requirement for an on-time forecast delivery for the purpose of the FIREX-AQ field
campaign. The 24-hour forecast simulations carried out in Figure 2 (c,f) each required nearly 3 hours of
supercomputer processing time; longer simulation periods were not possible within the operational window available
for forecasting.
The amalgamated observations and model pairs of vertical temperature profile data from 39 radiosonde sites in western
North America are shown in Figures 12 and 13. Improvements in the forecasted temperature vertical profile with
increasing forecast time are evident at 850 hPa in the 12th hour forecast (Figure 12) and at 925 and 1000 hPa in the
24th hour (Figure 13) forecast. The forecast simulation with aerosol feedbacks enabled also showed slight
improvements in the 10, and 50 hPa 12th hour temperatures and 50 hPa 24th hour temperatures, while 500 hPa 24th
hour temperature performance degraded slightly. There are larger differences between the 1000 hPa forecasts, though
these also have the least number of contributing stations (i.e. only those located close to sea-level contribute to the
lowest level temperature biases). Other levels of the atmosphere showed no statistically significant change at the 90%
confidence level in temperature profile forecast performance with the use of feedbacks.

## 3.2 Chemistry Evaluation

Chemistry forecast quality is usually evaluated using standard statistical metrics against hourly observations collected
from surface measurement stations. Both simulations' performance for ozone ($O_3$), nitrogen dioxide ($NO_2$) and
particulate matter with diameters less than 2.5 $\mu$m(PM2.5) were evaluated using hourly AIRNOW data (USA: AQS
network: https://www.epa.gov/aqs; Canada: NAPS network: http://maps-cartes.ec.gc.ca/rnspa-naps/data.aspx). The
summary performance metric scores for the two simulations grouped, according to contributing measurement network,
are shown in Table 2, with boldface values indicating the better score for the given simulation case. With respect to
this table, we note that:
(a) The feedback simulation generally outperforms the no-feedback simulation (more bold-face scores in the

"feedback" columns, with a few notable exceptions, discussed below).

(b) In some evaluation metrics, the feedback simulation showed substantial quantitative improvements over the no-

feedback simulation (e.g. feedback PM2.5 MB is reduced by over a factor of 3 relative to its no-feedback

counterpart over Western Canada, the region of wildfire activity).

(c) For cases when the no-feedback simulation outperforms the feedback simulation, the relative magnitude of the

performance difference is smaller than the feedback simulation's improvements (e.g. Western USA PM2.5 and

Western Canada $NO_2$ mean bias degradations of 9.0 and 27.8 percent relative to the PM2.5 improvement of a

factor of 3 noted above).





(d) Both simulations have negative mean biases for $O_3$ of -3.5 to -3.7 ppbv throughout the model domain, negative biases for PM2.5 in the Western USA, positive biases for PM2.5 in Western Canada, negative biases for $NO_2$ in Western Canada and positive biases for $NO_2$ in the Western USA.

One possible cause for the overall model biases noted in (d) may reside in inadequate chemical boundary conditions used for the forecasting setup. An unprecedented large high-resolution (2.5-km) model domain and on-time delivery of forecasts in support of FIREX-AQ put a significant constraint on available computational resources. 2.5-km chemical boundary conditions for these simulations were taken from seasonal climatologies from ECMWF global model analyses, rather than coarse resolution model simulations. During the time simulated, large forest fires occurred both within the domain (in northern Alberta and Saskatchewan in Canada), and outside of this model domain (in Alaska north of the Panhandle). These out-of-domain sources were thus not available as lateral boundary conditions on the 2.5-km domain, with possible impacts on model performance for the three species thus resulting in overall forecast biases in both the feedback and the no-feedback simulations, particularly in Western Canada.

The impact of lateral boundary conditions on model predictions can be seen when comparing MODIS retrievals of aerosol optical depth (AOD) with model predictions (Figure 14). AOD is a function of both the particle's abundance and optical properties, integrated throughout the vertical column. However, direct comparisons between satellite and model-predicted AOD values must be undertaken with some care, due to the nature of the satellite retrievals quality assurance and control procedures, the motion of the orbiting spacecraft, and the scan time of the instrument. For a polar-orbiting instrument such as MODIS, the time at which overpasses occur varies with location, and valid satellite retrievals may not occur when the location being scanned is obscured by clouds. Observed averages may be built up over multiple valid scans over time, but the number of valid scans contributing to the local average at any given location will vary, due to the time and space variation in cloud cover. Here, individual valid Collection 6.1 MODIS/Aqua (MYD04_L2 AOD_550_Dark_Target_Deep_Blue_Combined) 10 km resolution 550 nm AODs were matched in time and space to the nearest model 2.5-km grid cell and output frequency hour. Levy *et al.,* (2013) contains details on the MODIS combined AOD product. Both model and observed values were then locally averaged across the simulation period, to a common resolution 0.1° latitude-longitude grid co-located with the model grid, in order to generate the comparison shown in Figure 14. Individual pixels of this image thus incorporate a spatially varying number of values in the local averages, but these local averages are matched in time, space, and local averaging period. The region over which comparisons were made thus corresponds to the high resolution model domain, and white areas within the images correspond to regions where satellite data used in the averaging were excluded due to QA/QC constraints, such as the presence of clouds, surface ice, etc.

The MODIS-observed local average AOD values of Figure 14 (a,b) are generally higher than the model predictions (c,d), with the exception of the region co-located with large forest fires in Northern Alberta and Saskatchewan (Figure 14 (d), yellow regions). In addition, as noted above, other satellite imagery has shown the presence of several large forest fires occurring in Alaska, outside of the modelling domain, with smoke plumes extending from these sources down through the northern and western coastal boundaries of the model domain (yellow regions in Figure 14(a)). It is likely that at least some of the biases in PM2.5 and $NO_2$ and consequently in the production of secondary ozone, reflect the absence of these sources. Scatterplots of all paired AOD values in Figure 15 (a,b) and for the northern





portion of Alberta (Figure 15(c,d)) show that the overall negative bias is due to a large number of underestimated
"background" values of AOD in the model simulations, while in the immediate vicinity of the Alberta / Saskatchewan
forest fires, model values are considerably higher than observations.
The local model positive biases in AOD within Northern Alberta might partly be attributed to overestimates in emitted
particulate matter mass in the CFFEPS module, or from inaccurate treatment of fire plume centerline dispersion
downwind of primary emissions over large fire sources. Previous work with CFFEPS by Chen *et al. (*2019) for the
2017 fire season has shown similar $PM_{2.5}$ positive biases for western Canada, with MB of +5.8 µg m$^{-3}$ (88 stations)
and for Western USA with MB of +8.6 µg m$^{-3}$ (221 stations). These positive biases (Chen *et al.,* 2019) were higher
specific to sub-regions closer to areas of active fires (MB of +12 µg m$^{-3}$ for the sub-region including the provinces of
Alberta and British Columbia, and +29 µg m$^{-3}$ for the sub-region comprising the states of Idaho, Montana, Oregon
and Washington, respectively). Our analysis here and these earlier results suggest a positive bias in CFFEPS' PM2.5
emissions or insufficient dilution/vertical extent of the predicted fire plumes.
The local AOD positive biases could also be the result of the mixing state assumptions of the Mie code used here for
generating aerosol optical properties. These assumptions may also account for negative AOD biases over much of
the remainder of the model domain. We have used a mass-weighted homogeneous mixture approach, with the
complex refractive index values for the 8 particle species being calculated for pure water-dry component homogeneous
mixtures at each of the 12 particle size bins, followed by mass weighting to generate values for each of the model
components. As noted earlier, this overall negative bias of AOD predictions is a common problem in air-quality
models and may be due to assumptions regarding the model mixing state (Curci *et al.,* 2015). That comparison of
multiple mixing state assumptions on AOD with observations for European and North American modelling domains
(Curci *et al.,* 2015), showed a typical factor of two model under-prediction of 440 nm North American AOD across
all mixing state assumptions, with European AOD negative biases ranging from unbiased to a factor of 2. For the
latter group, those models employing an assumption of external mixing, with hygroscopic growth factors for sulphate
and nitrate assumed to be similar to those of sulphuric acid, had the highest AODs and hence closest values compared
to observations at 440nm. However, in that investigation, the latter method also sometimes resulted in AOD over-
predictions by a factor of 2 at 870 nm. These earlier findings along with overestimates at forest fire plumes with our
current homogeneous mixture approach at 550nm suggest that the hygroscopic growth may be overestimated for forest
fire particles, in turn overestimating forest fire AODs locally, while external mixing assumptions may be required to
improve model AOD performance elsewhere in the model domain.
**3.3 Model Evaluation Summary**
Overall, the incorporation of feedbacks in this study has resulted in improvements in weather and air-quality forecast
accuracy, albeit with some caveats. Weather forecast variables showed improvements at the 90% confidence level
for several fields, and vertical profiles showed improvements, particularly close to the surface, and with increasing
forecast lead time. Total precipitation scores also improved. A previously unexpected spin-up issue specific to fully
coupled models was noted: the impact of fully coupled particulate matter on cloud variables was sufficiently strong
that cloud field adjustment in the first 6 hours of the forecast was required prior to some weather forecast variable





improvements to be apparent (surface pressure, dewpoint temperature, sea-level pressure). While the current forecast
cycling duration was constrained by operational requirements, this suggests that forecast cycling should include both
air-quality and meteorological variables during fully-coupled forecast spin-up periods. That is, the model tracer
concentrations 6 hours prior to the current forecast start-up could also be used during the initial meteorological spin-
up period, thus allowing chemistry and cloud formation to spin-up simultaneously. Scores for surface PM2.5, NO$_2$,
and O$_3$ also generally improved with the incorporation of feedbacks, with some metrics showing large improvements.
In comparison to satellite-based AOD values, the current model's AOD values were generally biased low, with
exceptions being in the regions of Alberta and Saskatchewan with active forest fires where AOD was biased high.
The latter comparison also showed that large fires off-domain in Alaska likely had a large impact on AODs in the
eastern and northern section of the model domain due to missing boundary condition contributions. These sources
were missing due to operational limitations in the model simulations shown here.
**4 Effects of Feedbacks on Selected Simulation-Period Average Variables**
In this section, we compare time averages of the entire study period for the two simulations, both at the surface and in
vertical cross-sections through the model domain, to illustrate some of the changes in both weather and air-quality
associated with the incorporation of feedbacks. We have found differences at greater than 90% confidence between
the predicted meteorological and chemical forecasts in the vicinity of the Alberta/Saskatchewan forest fires, as well
as in contrasting changes between land and sea. We note again here that the "no-feedback" simulation makes use of
time and spatially invariant aerosol CCN and optical properties, within the meteorological portion of the model. The
comparisons thus show the differences associated with the use of climatological constant aerosol properties, and the
fully coupled model-generated aerosols.
As in the meteorological evaluation, we have made use of 90% confidence levels in order to gauge the level of
significance of the differences between the feedback and no-feedback simulations in the following analysis. At each
model grid cell the values of the standard deviation about the mean for each respective simulation was calculated.
The difference between the means becomes significant at a given confidence level c if the regions defined by $M_f \pm$
$z^* \frac{\sigma_f}{\sqrt{N}}$ an$M_{nf} \pm z^* \frac{\sigma_{nf}}{\sqrt{N}}$d do not overlap, where N is the number of points averaged, $M_f$ is the feedback mean value, $M_{nf}$
is the no-feedback mean value, $\sigma_f$ and $\sigma_{nf}$ are the corresponding standard deviation, and $z^*$ is the value of the $\sqrt{c}$
percentile point for the fractional confidence interval $c$ of the normal distribution ($z^* = 1.645$ at $c = 0.90$). Grid cell
values where this overlap does not occur (i.e. where the mean values differ at or above the 90% confidence level) may
be defined via the following equation[1]:

---

[1] Note that for cases where $M_{nf} > M_f$, significance at the confidence level associated with $z^*$ occurs when the range
of standard errors about the mean do not overlap, ie. $M_{nf} - z^* \frac{\sigma_{nf}}{\sqrt{N}} > M_f + z^* \frac{\sigma_f}{\sqrt{N}}$, or

$\left(M_{nf} - M_f\right)/\left(\frac{z^*}{\sqrt{N}}\left(\sigma_{nf} + \sigma_f\right)\right) > 1$. Similarly, for cases where $M_f > M_{nf}$, significance at the confidence level

associated with z* occurs when $\left(M_f - M_{nf}\right)/\left(\frac{z^*}{\sqrt{N}}\left(\sigma_{nf} + \sigma_f\right)\right) > 1$. Equation (1) may thus be used to describe both
conditions.





$$\frac{|M_{nf}-M_f|}{\frac{z^*}{\sqrt{N}}(\sigma_f+\sigma_{nf})} > 1 \qquad (1)$$
The differences in the mean grid cell values between the simulations for which the above quantity is greater than unity
thus differ at or greater than the 90% confidence level. Differences in the mean values, as well as the value of the
above ratio, are thus reported in the following section.
**4.1 Effects of Feedbacks on Time-Averaged Meteorology**
The feedback – no-feedback differences in the simulation-period average cloud droplet number density (number kg⁻¹
of air) and mass density (g water kg⁻¹ of air) along centred cross-sections spanning the length and width of the model
domain are shown in Figure 16 (the locations of the cross-sections are shown in Figure 1). The "Ocean", "Land", and
"Forest Fire" regions identified are with reference to the approximate locations of these features along these cross-
sections. Figure 16 also shows the confidence ratio values as described above – regions where the predicted mean
values differ at or above the 90% confidence level are shown in red, while those differences below the 90% confidence
interval are shown in blue. Feedbacks increase the cloud droplet number density over the northern part of the domain,
including the region impacted by the Alberta/Saskatchewan forest fires, from the surface up to about 600 mb (roughly
equivalent to hybrid level 0.600), and decrease further aloft and along the length of the model domain into the western
USA (Figure 16(a)). Cloud droplet numbers also decrease over the ocean, but increase eastwards over the land (Figure
16(b)). The latter is unrelated to the forest fires; this is an indication that the modelled aerosol number concentration
over the ocean is much lower than the single climatological aerosol population assumed in the no-feedback run,
resulting in lower cloud droplet number concentrations. In both cases, the differences are significant at the 90%
confidence level from the surface up to hybrid level 0.87 and in isolated regions at hybrid level 0.550 along the south
to north cross-section (Figure 16(c)), and over the ocean in the west to east cross-section (Figure 16(d)). Higher-than-
climatology aerosol loadings, a large portion of which are due to the forest fires, resulted increased cloud droplet
number densities in the lower troposphere, while decreasing them in the mid-to-upper troposphere. This impact of
feedbacks is in accord with the satellite observations of Saponaro *et al. (*2017), and was also seen in Takeishi *et al.*
(2020). In contrast, cloud droplet mass density(i.e. cloud liquid water content) largely decreases across the domain
along the north-south cross-section (Figure 16(e)), as well as over the ocean, with a varying pattern over the land in
the east-west cross-section (Figure 16(f)). The magnitudes and significance levels for the average change in cloud
droplet mass are lower than for cloud droplet number, with the most significant differences occurring over the ocean
(Figure 16(g,h)).
Consistent with the cloud droplet number changes, rain droplet numbers and mass mixing ratios increase with the
feedback simulation over both the forest region impacted by the forest fires (Figure 17(a,e)) and over the ocean (Figure
17(b,f)), with a varying impact over the land and more distant from the forest fire sources (Figure 17(f)). The changes
are significant at the 90% confidence level for rain droplet number in these regions (compare Figure 17(a) with 17(c);
17(b) with 17(d)), while the rain droplet mass changes approach but are below the 90% confidence level.
These results suggest that relative to the no-feedback simulation, which employs climatological aerosol CCN
properties, the AIE in the feedback simulation is causing significant change in hydrometeor numbers, and a less





significant increase in hydrometeor mass.    In the forest fire-impacted region, the ADE and AIE in the feedback
simulation significantly increase the number of cloud droplets near the surface and decrease the number of cloud
droplets in the middle to upper troposphere (Figure 16(a,c)). The rain drop number in the middle troposphere (Figure
17(a,c)) also increases significantly between hybrid levels 0.90 to 0.70 (Figure 17(e,g)).  Near-surface rain drop
number and rain drop mass differences throughout the cross sections (Figure 17(e,f)) fall below the 90% confidence
level (Figure 17(g,h).
Over the oceans, water droplet number and mass both decrease (Figure 16(b,f)), and raindrop number and mass
increase (Figure 17(b,f)); more atmospheric water is converted to rain drops as a result of the feedbacks, relative to
the climatology in the no-feedback simulation.  However, these changes are more significant aloft than at the surface,
with the difference in both rain drop number and mass falling below the 90% confidence level near the surface.  We
interpret these changes as a shift in over-ocean liquid hydrometeor numbers and to a lesser degree the water mass aloft
from cloud droplets to rain drops due to the AIE in the feedback setup relative to the climatology of the no-feedback
simulation.  The changes occur at the 90% confidence level aloft, but the near-surface changes are smaller and are
usually below the 90% confidence level.
Differences in the average precipitation flux and the confidence ratio values are shown in Figure 18.  Changes in
average precipitation (Figure 18(a)) appear random, though locally these differences are significant at the 90%
confidence level (Figure 18(b)).  Both the magnitude of the differences and the frequency in their reaching the 90%
confidence level increase westwards.  Given the local and episodic nature of rainfall events the high level of
significance in this case probably results from the presence or absence of individual rainfall events between the two
simulations affecting the local average and standard deviations.
Several systematic changes in the average values of the model's meteorological output fields were noted due to the
use of feedbacks relative to aerosol property climatologies (Figure 19), although all fall below the 90% confidence
level for the difference in the mean values between the two simulations (Figure 20). Surface air temperature generally
increased (Figure 19(b)) though less so in the region most affected by forest fires, dewpoint temperature decreased
(Figure 19(c)) implying a decrease in relative humidity with feedbacks.  Surface pressure increased, particularly in
the region downwind of the Alberta / Saskatchewan fires (Figure 19(d)).  Planetary boundary layer height increased
over the land (Figure 19(e)), consistent with decreased atmospheric stability.  The friction velocity also increased with
the use of feedbacks (Figure 19(f)); this is consistent with a decrease in stability and an increase in turbulent energy
The air temperature increases are limited to the lowest part of the atmosphere (Figure 21 (a,b)), usually below hybrid
level 0.914 (approximately 1km above the surface).  The feedbacks decrease temperatures between hybrid levels 0.914
and 0.721.  Feedbacks thus increase or do not affect temperatures near the surface, and decrease temperatures in the
lower free Troposphere.  However, all of these features, despite their large geographic range, fall below the 90%
confidence level, reflecting the large variability in surface temperatures contained within each simulation.  Longer
time simulations than carried out here are required in order to improve confidence in the temperature predictions
across all forecast hours.  However, these results, particularly for surface temperature, may be contrasted with Figures
6(a), 12 and 13, which suggests that temperature differences rise above the 90% confidence interval at specific forecast
times at surface and upper atmosphere measurement sites.





### 4.2 Effects of Feedbacks on Time-Averaged Chemistry

In the previous meteorological impacts section, changes in aerosol loading relative to the climatology, dominated by forest fires, were shown to have a significant impact on cloud formation and atmospheric temperatures through ADE and AIE. These might be expected to in turn influence and be influenced by particulate matter emitted by the forest fires, with the plume rise of the forest fires dependent on the meteorological changes. Air temperatures increase slightly in the model surface layer (Figure 19(b), +0.01 to +0.05 ℃) but decrease at greater magnitudes through the rest of the lower Troposphere (hybrid level 0.980 up to 0.667, Figure 21), with a maximum decrease of -0.5℃ between hybrid levels 0.871 and 0.824. The increase in air temperatures near the surface and decreases aloft (a stronger negative vertical gradient in temperature) implies a decrease in atmospheric stability associated with feedbacks, given that the overall temperature gradient from the surface has become more negative. This is echoed by the response of the concentration fields to the near-surface stability change, as can be seen through comparisons of the PM2.5, $NO_2$ and $O_3$ surface concentrations changes (Figure 22) and as vertical cross-sections (Figures 23, 24, 25), respectively.

For all three surface fields, changes above the 90% confidence level occur near the forest fires themselves (red regions, near top of model domain, Figure 22(a,b,c)). The differences in particulate matter concentrations are also significant at the 90% confidence level throughout the model domain (Figure 22(a)). Note that while the PM2.5 mean values are significantly different at the 90% confidence level throughout the model domain, the magnitude of those differences are sometimes small, particularly in the upper atmosphere, where the aerosol concentrations are relatively small. However, the regions with the larger magnitude regional differences in PM2.5 concentrations also occur at greater than the 90% confidence level (compare spatial locations of coloured regions in Figure 22(a,b) to red regions in Figure 22(c,d)).

Near-surface PM2.5 decreases in the regions downwind of the forest fires (Figure 22(a), Figure 23(a), note the large blue region and more intense blue region near surface in 23(a)), suggesting less PM2.5 mass is present near the surface due to the feedbacks. This could reflect a change in injection height of the plumes in addition to other transport changes associated with the decrease in atmospheric stability. Lower troposphere decreases in PM2.5 over the ocean and increases over the land at or greater than the 90% confidence level may also be seen (Figure 23(b,d)).

Feedbacks result in an increase in near-surface $NO_2$ in several inland urban centers and less $NO_2$ at surface level downwind (Figure 22(b), though these differences are only significant at the 90% confidence level within the forest fire plumes (Figure 24(a,c)). Ocean versus land $NO_2$ differences remain below the 90% confidence level.

Feedbacks decreased surface $O_3$ near the forest fires (Figure 22(c), Figure 25(a)), while decreasing increasing $O_3$ aloft. The forest fires are also the only area where the differences in between mean ozone forecasts have greater than the 90% confidence.

Overall, the most significant effects of the feedbacks were: (1) changes in PM2.5 concentrations throughout the model domain, and (2) changes in $NO_2$ and $O_3$ within the forest fire plumes.

The feedback-induced changes in primary and secondary pollutants in the forest fire regions are consistent with the decrease in atmospheric stability noted above – a greater proportion of the primary particulate matter and $NO_2$ resulting from near-surface forest fire emissions of NO are carried upwards with the addition of feedbacks. The decrease in surface ozone and increase further aloft in the fire region (Figure 25(a)) spatially matches the changes in $NO_2$ (Figure





23(a)) – this implies that the changes associated with feedbacks occur in NOx-limited environments, i.e., with
relatively high VOC/NOx ratios. In such environments, decreases in NOx emissions may lead to decreases in the rate
of secondary $O_3$ formation.
Our analysis thus suggests enhanced upward transport occurs in forest fire plumes due to feedbacks, and that this
transport is linked to feedback-induced: (1) decreases in local atmospheric stability (Figure 21(a)); (2) increases in
cloud droplet numbers near the surface (Figure 16(a)); and (3) increases in rain drop numbers aloft (Figure 17(a)).
This combination suggests the presence of an AIE feedback loop – decreased stability results in higher forest fire
plume rise, in turn lofting proportionally more particles to higher levels in the atmosphere where they may act as cloud
condensation nuclei, increasing cloud droplets aloft (Figure 16(a)). This in turn results in increased lower middle
troposphere cooling, through the 1[st] AIE (increase in cloud droplet numbers aloft leading to increased cloud albedo
and cooling of the atmosphere below the cloud tops) while the corresponding decreases in particles and cloud
condensation nuclei at lower levels results in a smaller near-surface impact on the AIE and ADE, hence relatively
minor changes on near-surface temperatures (Figure 21(a)). This combination maintains a feedback-induced less
stable temperature gradient, relative to the no-feedback simulation employing aerosol property climatologies.
Similarly, over the oceans, the feedback-induced decrease in surface PM2.5 (Figure 23(a)) is accompanied by lower-
middle troposphere cooling (Figure 21(b), note the dark blue band over ocean between hybrid levels 0.914 and 0.824),
implying a decrease in stability. While aerosol increases aloft are not large, the lower level PM2.5 act as CCN,
resulting in increased cloud formation and convection, in turn increasing cloud droplet and rain drop numbers aloft,
which through the 1[st] AIE, maintains the slightly less stable temperature profile. We acknowledge that these changes
in temperature fall below the 90% confidence level for the averages over all times, though note that differences in
mean bias relative to observations for the two simulations became significantly different at specific times of day in
the forecasts (Figure 6(a)), hours 3, 6, 15 and 18, corresponding to 15, 18, 3 and 6 UT, or 9 AM, 12 noon, 9 PM, and
midnight MDT), implying that the temperature changes at these specific times reach a higher level of significance.
**4.3 Summary, Differences in Forecast Simulation-Period Averages**
Relative to the no-feedback simulation employing an aerosol climatology, the AIE feedback as simulated here is
associated with decreases in stability over both ocean and forest-fire influenced land areas. Over oceans, near-surface
particulate matter is removed as cloud condensation nuclei, resulting in increased cloud droplet numbers, maintaining
the temperature gradient through the 1[st] aerosol indirect effect. In the vicinity of forest fires, decreases in stability
result in increased transport of PM2.5 aloft, increasing the availability of cloud condensation nuclei aloft, increasing
cloud droplet numbers aloft, hence also maintaining the increased temperature gradient through the 1[st] aerosol indirect
effect. We note that the ADE may also play a weak role, particularly in the southern part of the domain, where lower
atmosphere temperature gradient increases are not accompanied by significant changes in cloud droplet numbers
(Figure 16(a), southern half of the cross-section), but are accompanied by significant though small magnitude increases
in PM2.5 in the lower atmosphere (Figure 23(a), southern half of cross-section), and temperature profile changes
(Figure 21) below the 90% confidence level.



## 5 Conclusions

The work carried out here suggests that the answers to our two research questions ("Can fully coupled models improve both air-quality and meteorological forecasts?" and "Are the changes in forest fire forecasts associated with implementing forest fire emissions within a fully coupled model sufficient to significantly perturb weather and meteorology?") are both a qualified "yes".

The simulations analyzed here were conducted in preparation for an experimental forecast carried out as part of the FIREX-AQ campaign, and hence were limited by operational time constraints to a sequence of nested 24-hour forecasts. However, the high resolution domain size employed was sufficiently large to result in improvements in weather forecast performance for both surface and profile variables at or above the 90% confidence level. Improvements in model performance for PM2.5, $NO_2$ and $O_3$ were also found, across most statistical measures. The differences between feedback and no-feedback simulations occurred at or above the 90% confidence level throughout the model domain for PM2.5, though were limited at the 90% confidence level for $NO_2$ and $O_3$ to the immediate vicinity of the forest fires. There, increased vertical transport associated with feedbacks lowered near-surface $NO_2$ concentrations and increased them aloft, resulting in reduced surface $O_3$ in the NOx-limited regions of the forest fire plume.

The simulations suggest that the homogeneous mixture approach for aerosol optical properties results in a general under-prediction of aerosol optical depths, in accord with Curci *et al.* (2015), and external mixture approaches are recommended in further study. However, this general negative bias in simulated AOD is locally offset by positive biases in the vicinity of forest fires. This suggests that forest fire plumes have significantly different optical properties, and may be less hygroscopic than industrial aerosols of comparable size. Special / separate treatment of forest fire CCN and optical properties are therefore also recommended in future work.

Fully coupling forest fire plume rise calculations with the weather parameters was shown to have a significant impact on the height of primary pollutants reached by forest fires, the formation of near-surface ozone near the forest fires, and on particulate matter throughout much of the three-dimensional model domain. These changes were largely driven by the AIE, which maintains an increased temperature gradient (reduced stability) over the forest-fire-influenced and oceanic portions of the region studied. Weak evidence for the influence of the ADE was shown in the southern part of the domain, where increases in particulate matter were also accompanied by decreases in stability between the surface and the lower-middle troposphere (the differences were at a lower than 90% confidence level for these comparisons of temperatures averaged over all model times).

Relative to the no-feedback aerosol climatology for CCN and aerosol optical properties, the simulations carried out here suggested that in the vicinity of forest fires feedbacks significantly increase cloud droplet near the surface, increase cloud droplet number aloft, and significantly increase rain drop number densities aloft, relative to forecasts driven by climatological aerosol properties. Over the oceans, feedbacks decreased cloud droplet number density and increased rain drop number density aloft. Cloud droplet mass increased to a lesser degree (with smaller regions above the 90% confidence level), as did rain drop mass (the mean differences for which for the most part remained below the 90% confidence level). This provides some evidence for a shift in atmospheric water mass associated with feedbacks from cloud water to rain over the oceans relative to the no-feedback climatology, though this shift occurred




largely within the variability of the cloud fields within each simulation. Longer simulations may be needed to achieve
higher confidence in this finding.

**Data Availability**
The datasets used here for model evaluation are available from the publicly accessible websites (AQS network)
https://www.epa.gov/aqs and (NAPS network) http://maps-cartes.ec.gc.ca/rnspa-naps/data.aspx.
**Author Contribution**
PAM: experiment design, conceptualization, analysis, writing of manuscript drafts; AA: model code and run script
design and implementation, statistical analysis of model results, model analysis graphics; JC: forest fire emissions
processing system design and coding, manuscript contributions, draft review and assistance; BP: forecast system
simulations and design; Wanmin Gong[1]: indirect effect updates, advice on P3 implmentation, manuscript
contributions, manuscript review; CStroud: code version contributions, manuscript review; CSioris: AOD analysis,
manuscript contributions and review; KS: forest fire emissions processing system design and coding, manuscript
contributions and review; PC: forecast system simulations and design; JZ: emissions processing and input field
assistance, manuscript review and contributions; J.M.: indirect effect updates and advice on implementing AIE in the
P3 scheme, manuscript review and contributions.

**Competing Interests**
The authors declare that they have no conflict of interest.

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





**Tables:**

| Event Forecast | Event Observed | |
|---|---|---|
| | Yes | No |
| Yes | A | B |
| No | C | D |

**Table 1. Event versus non-event contingency table. A = number of events forecast and observed; B=number of events forecast but not observed; C=number of events observed but not forecast; D = number of cases where events were neither forecast nor observed.**





**Table 2: Summary performance metrics for ozone, nitrogen dioxide, and PM2.5. Bold-face indicates the simulation with**
**the better performance score for the given metric, chemical species and sub-region, italics indicate a tied score, and regular**
**ont the simulation with the lower performance score. FO2: fraction of scores within a factor of 2. MB: Mean Bias. MGE:**
**Mean Gross Error. R: Correlation Coefficient. RMSE: Root Mean Square Error. COE: Coefficient of Error. IOA:**
**Index of Agreement.**

| Chemical | Region | Simulation | FO2 | MB | MGE | NMGE | R | RMSE | COE | IOA |
|---|---|---|---|---|---|---|---|---|---|---|
| **PM2.5** | Western Canada | No Feedback | 0.451 | 0.777 | 4.335 | 0.878 | **0.278** | 7.642 | -0.669 | 0.165 |
| | | Feedback | **0.453** | **0.236** | **4.215** | **0.829** | 0.219 | **6.976** | **-0.534** | **0.233** |
| | Western USA | No Feedback | **0.536** | **-1.639** | 3.793 | 0.605 | 0.358 | 6.061 | -0.166 | 0.417 |
| | | Feedback | 0.524 | -1.786 | **3.773** | **0.602** | **0.361** | **5.978** | **-0.162** | **0.419** |
| **O₃** | Western Canada | No Feedback | 0.773 | -3.683 | 7.76 | 0.347 | 0.634 | 9.996 | 0.138 | 0.569 |
| | | Feedback | **0.78** | **-3.553** | **7.693** | **0.344** | **0.635** | **9.854** | **0.145** | **0.573** |
| | Western USA | No Feedback | 0.879 | -3.584 | 9.667 | 0.257 | *0.763* | 12.534 | 0.319 | 0.66 |
| | | Feedback | **0.881** | **-3.456** | **9.607** | **0.256** | *0.763* | **12.458** | **0.322** | **0.661** |
| **NO₂** | Western Canada | No Feedback | **0.528** | **-0.417** | 2.29 | 0.594 | **0.519** | 3.353 | 0.053 | *0.527* |
| | | Feedback | 0.52 | -0.533 | **2.274** | **0.591** | 0.513 | **3.314** | **0.055** | *0.527* |
| | Western USA | No Feedback | **0.428** | 0.669 | 2.278 | 0.759 | **0.585** | 3.917 | -0.161 | 0.419 |
| | | Feedback | 0.427 | **0.578** | **2.24** | **0.746** | 0.581 | **3.858** | **-0.141** | **0.429** |




**Figures:**



Figure 1: GEM-MACH domains: (a) Operational GEM 10km resolution forecast domain. (b) Experimental 2.5-km grid cell size forecast domain used here. Red lines indicate locations of illustrative South to North and West to East cross-sections appearing in subsequent analysis in the text.





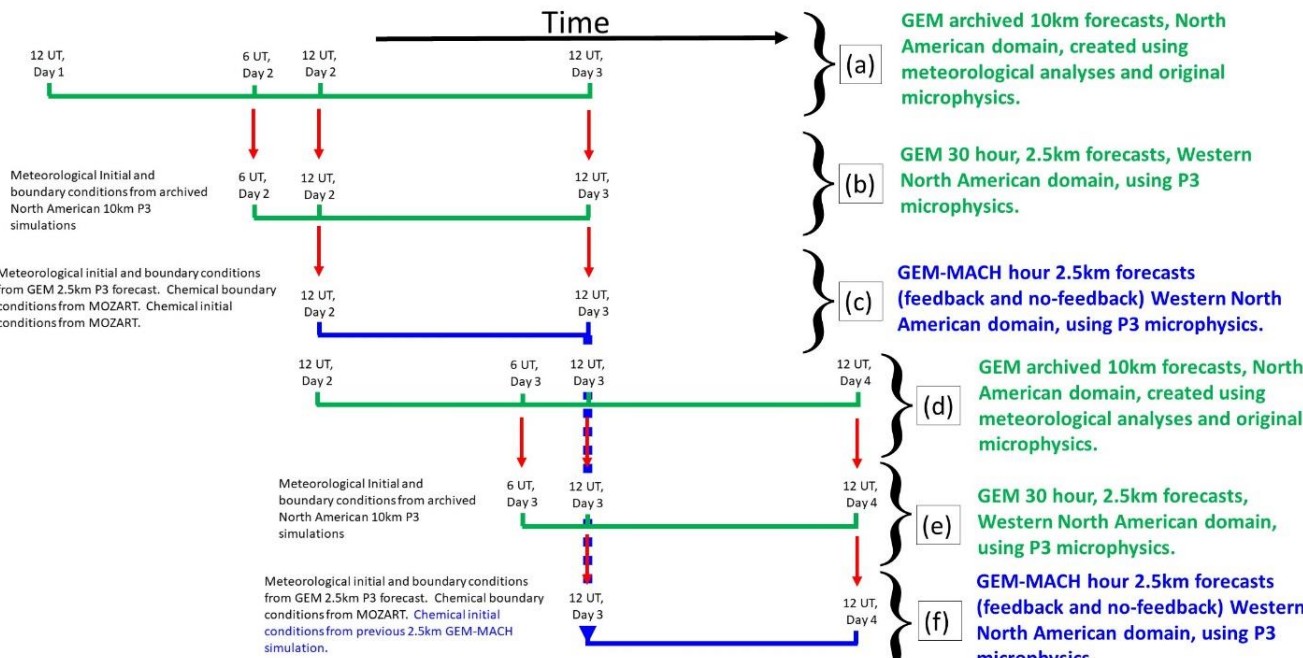

**Figure 2: Example time sequencing of model simulations used to generate the 2.5-km GEM-MACH simulations carried out here. Green lines and print indicate GEM (weather forecast only) simulations), blue lines and print indicate 2.5-km GEM-MACH simulations. Steps (a) through (f) illustrate the sequence of forecasts used to generate two consecutive days of 2.5km GEM-MACH simulations.**





**Figure 3:** **Process comparison between original (CFFEPSv2.03, left) and on-line (CFFEPSv4.0, right) forest fire emissions and vertical plume distribution algorithms.**

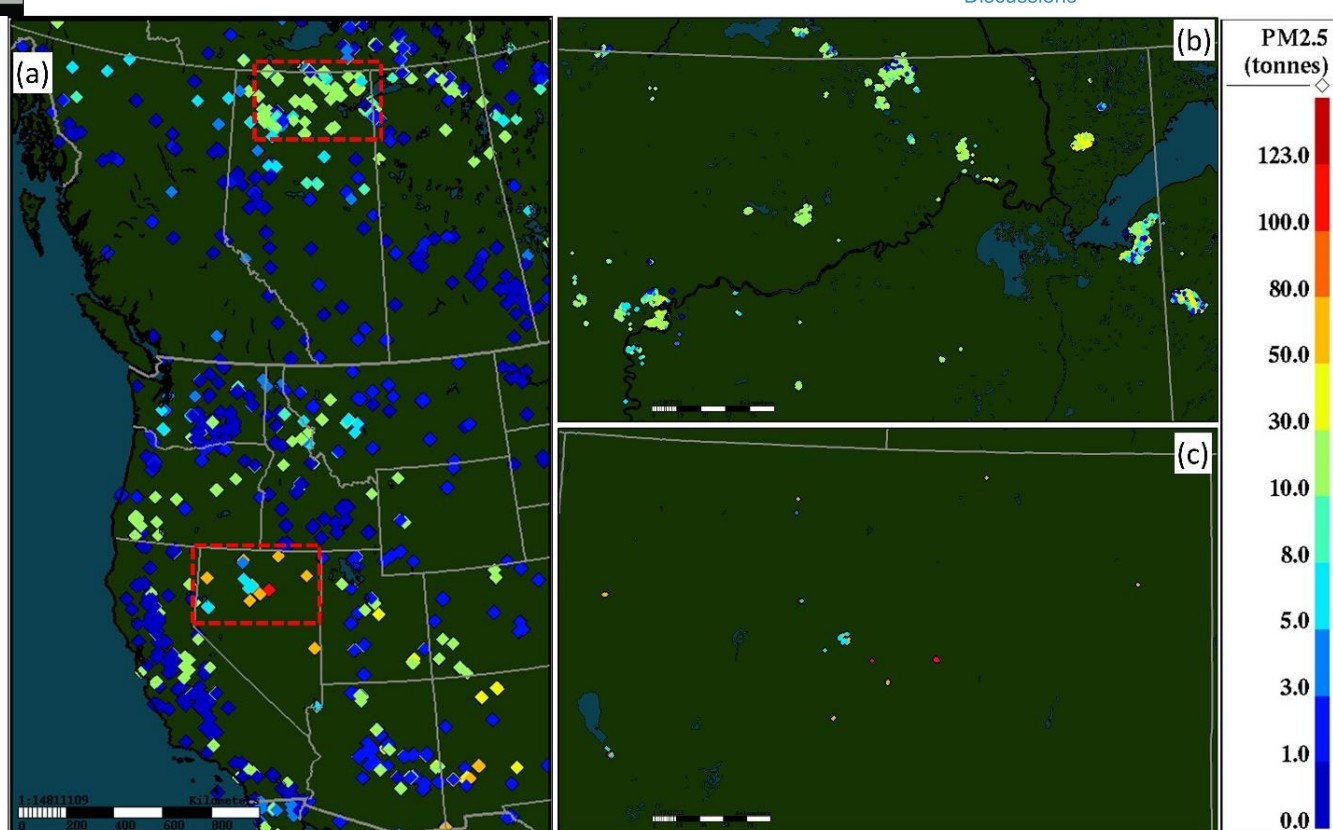

**Figure 4: Hotspot locations during the study period, colour-coded by daily total tonnes PM2.5 emitted. (a) Entire model 2.5-km domain, with northern Alberta and northern Nevada sub-regions as red dashed boxes; (b) northern Alberta zoom, with smaller symbols for individual hotspots showing the large fire regions; (c) northern Nevada zoom, to the same scale as (b), showing isolated hotspots with high emissions.**



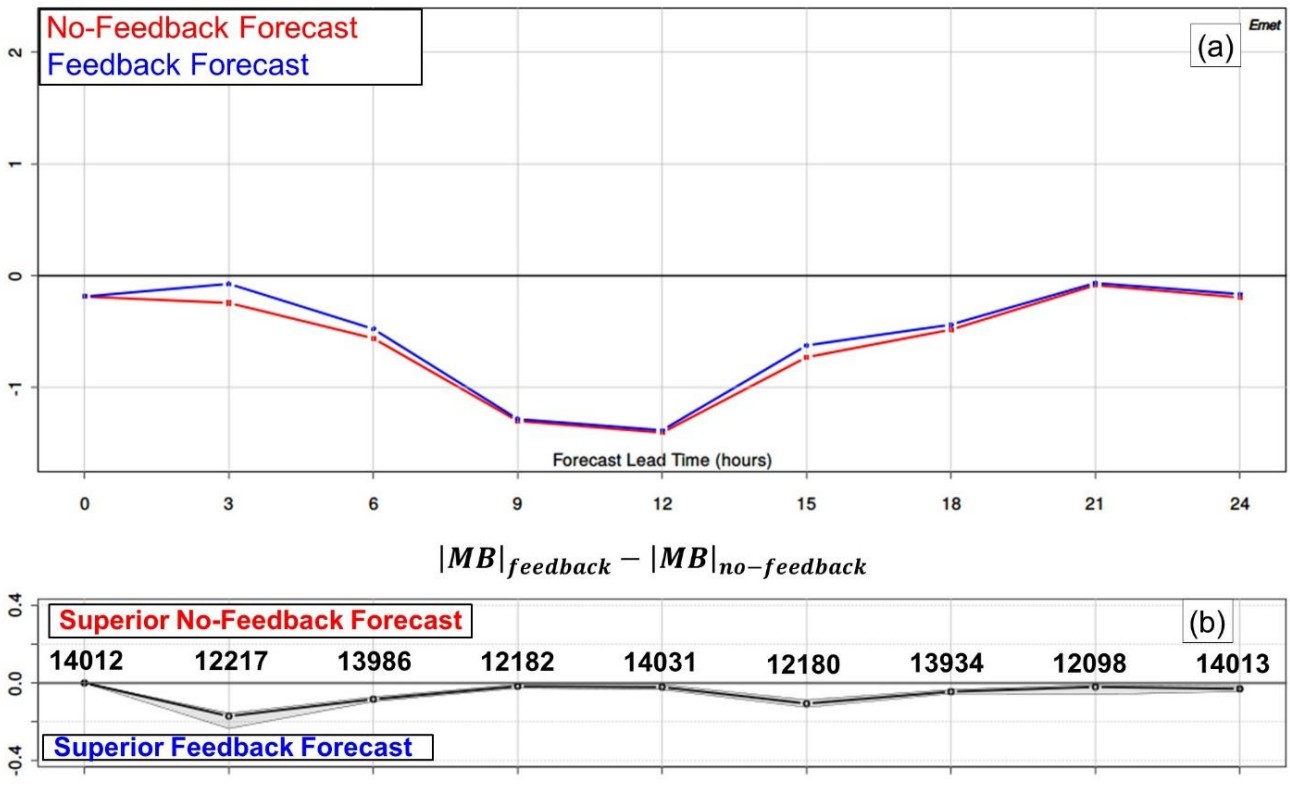

**Figure 5:** **Mean bias in surface temperature (°C) at forecast hours starting at 0 UT. (a) Red line: no-feedback forecast values; blue line: feedback forecast values. (b) Difference in absolute value of mean bias between the two forecasts $\left(|MB|_{feedback} - |MB|_{no-feedback}\right)$, with the region *below* 90% confidence level shown shaded grey. Mean values above above/below the '0' line, and outside of the shaded region thus indicate differences in the mean between the two forecasts which differ at or above the 90% confidence level. Values of the difference which appear below/above the zero line and outside of the grey area thus indicate superior domain average performance for the feedback/no-feedback forecasts at each of the 3-hourly intervals, respectively. Numbers appearing above the metric differences are the number of observations contributing to the calculated metrics.**




$$|MB|_{feedback} - |MB|_{no-feedback}$$

(a)

Superior No-Feedback Forecast

14012  12217  13986  12182  14031  12180  13934  12098  14013

Superior Feedback Forecast

$$MAE_{feedback} - MAE_{no-feedback}$$

(b)

Superior No-Feedback Forecast

Superior Feedback Forecast

$$RMSE_{feedback} - RMSE_{no-feedback}$$

(c)

Superior No-Feedback Forecast

(very slight)

Superior Feedback Forecast

$$R_{no-feedback} - R_{feedback}$$

(d)

Superior No-Feedback Forecast

Superior Feedback Forecast

Forecast Hour

**Figure 6: Summary meteorological performance comparison for surface temperature (C). (a) mean bias, (b) mean absolute error, (c) root mean square error and (d) Pearson correlation coefficient. 90% confidence level shown in grey. Numbers appearing above the absolute mean bias differences are the number of stations contributing to the calculated metrics.**





**Figure 7: Summary meteorological performance comparison for surface pressure (hPa). (a) mean bias, (b) mean absolute error, (c) root mean square error, (d) Pearson correlation coefficient, and (e) standard deviation. 90% confidence level shown in grey. Numbers appearing above the absolute mean bias differences are the number of stations contributing to the calculated metrics.**





**Figure 8: Summary meteorological performance comparison for dewpoint temperature (C). (a) mean bias, (b) mean absolute error, (c) root mean square error, (d) Pearson correlation coefficient, and (e) standard deviation. 90% confidence level shown in grey. Numbers appearing above the absolute mean bias differences are the number of stations contributing to the calculated metrics.**



**Figure 9:** Summary meteorological performance comparison for 10m windspeed (m s$^{-1}$). (a) mean bias, (b) mean absolute error, (c) root mean square error, (d) Pearson correlation coefficient, and (e) standard deviation. 90% confidence level shown in grey. Numbers appearing above the absolute mean bias differences are the number of stations contributing to the calculated metrics.





**Figure 10:** Summary meteorological performance comparison for sea-level pressure (hPa). (a) mean bias, (b) mean absolute error, (c) root mean square error, (d) Pearson correlation coefficient, and (e) standard deviation. 90% confidence level shown in grey. Numbers appearing above the absolute mean bias differences are the number of stations contributing to the calculated metrics.



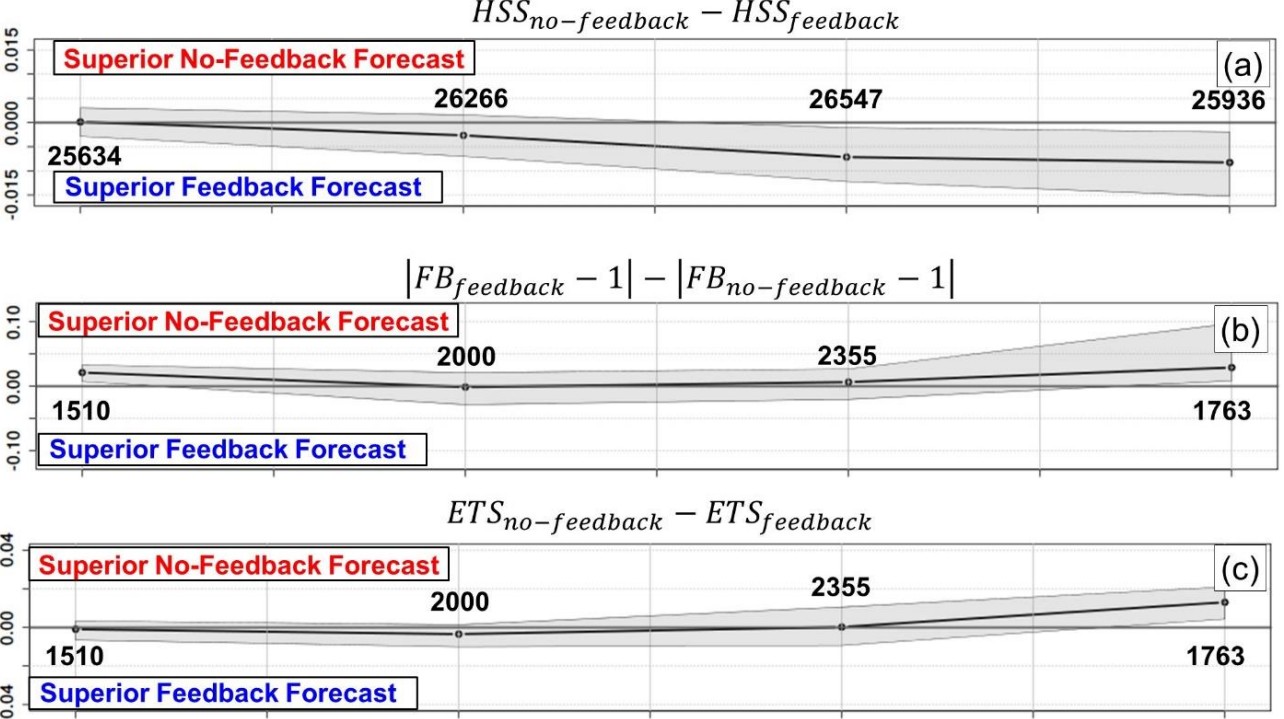

**Figure 11: Precipitation performance evaluation (mm precipitation). (a) Heike skill score of 6-hour accumulated precipitation (No-Feedback – Feedback). (b) Frequency bias index of 6-hour accumulated precipitation (threshold of 2 mm, No-Feedback – Feedback). (c) Equitable Threat Score of 6-hour accumulated precipitation (threshold of 2 mm, No-Feedback – Feedback).**





**Figure 12: Forecast hour 12 (0 UT) summary upper air temperature performance comparison for air temperature (mean bias, C). (a) Difference in absolute value of mean bias in temperature, (feedback forecast – no-feedback forecast). Grey regions represent 90% confidence levels, blue symbols: pressure levels at which the feedback mean bias outperforms the no-feedback mean-bias at > 90% confidence. Red symbols: pressure levels at which the no-feedback mean bias outperforms the feedback mean bias at > 90% confidence. 90% confidence level shown in grey. (b) Mean bias in upper air temperature for feedback (blue) and no-feedback (red) (C). Numbered values on the profiles indicate the number of observed data-model pairs at each pressure level.**





**Figure 13: Forecast hour 24 (12 UT) summary upper air temperature performance comparison for air temperature (mean bias, C). (a,b) as in Figure 12.**





**Figure 14: 550nm AOD comparison. (a) MODIS observations spatiotemporally averaged and sampled over the model domain and forecast duration and (b) zoom-in on the province of Alberta. (c,d) GEM-MACH nearest output hour AOD values corresponding to satellite sensor data availability for the same regions.**





**Figure 15:** **Scatterplots comparing model and observed AOD values. (a|) All model-observation pairs. (b) As in (a), rescaled to focus on AODs in the 0-3 range. (c) Model-observation pairs in the region bounded by 56N, 60N, -110W and -120 W (northern Alberta). (d) As in (c), rescaled to focus on AODs in the 0-3 range.**



**Figure 16. (a,b) Difference in mean (Feedback – No-Feedback) cloud droplet number simulations along south to north and east to west cross-sections through the middle of the model domain. (c,d) Corresponding significance level of mean cloud droplet number**

**differences using the confidence ratio defined in equation (1) – red areas indicate ratio values greater than unity, i.e., significance at or above the 90% confidence level. (e,f) Difference in mean cloud droplet mass (g kg⁻¹) (g,h) Corresponding significance level of mean cloud droplet mass difference.** *Note: the vertical axis in hybrid coordinates does not show all model levels for clarity; the model has much finer resolution in the lower part of the atmosphere than shown, and the portion of the vertical domain shown encompasses only the lower half of the levels in the model.*

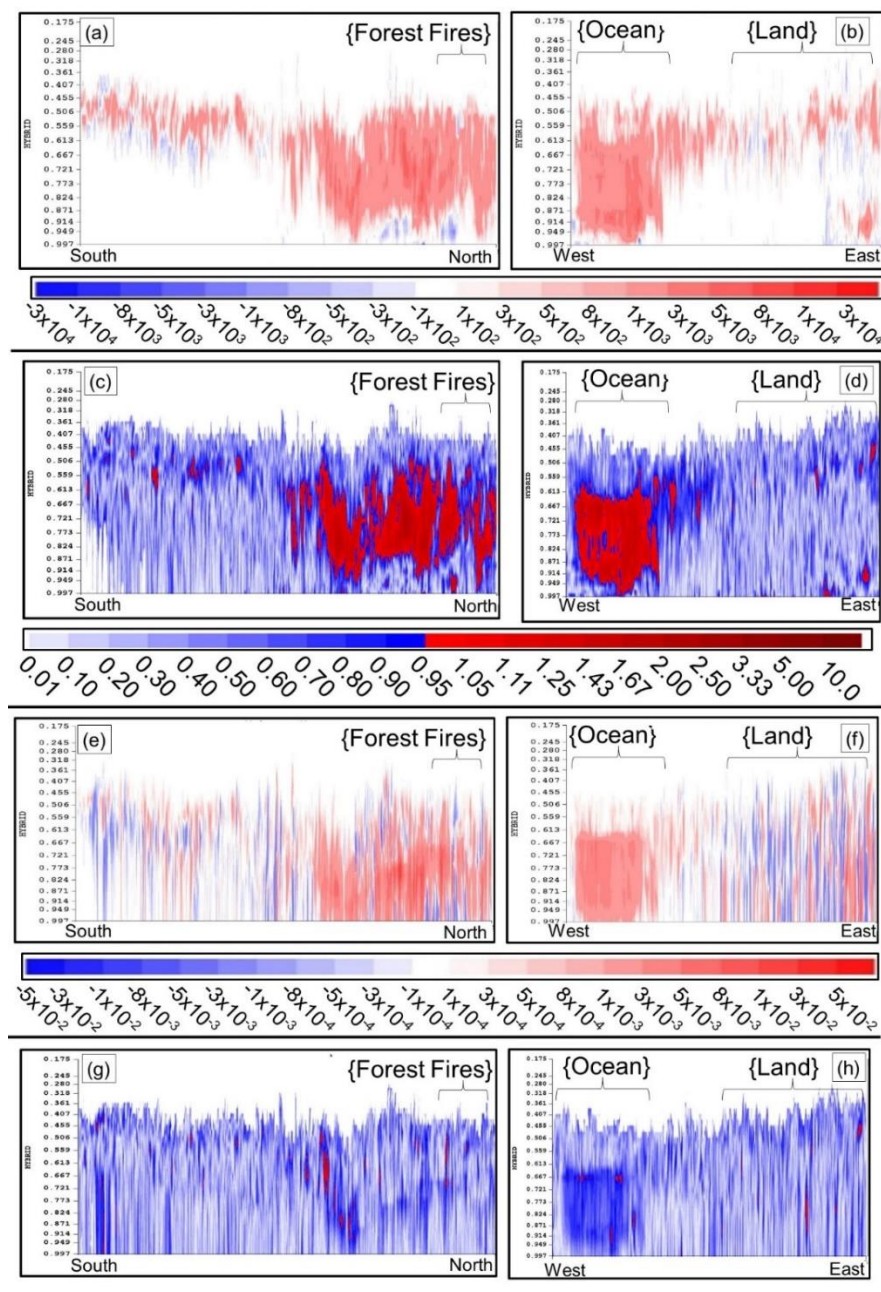

**Figure 17. (a,b) Difference in mean (Feedback – No-Feedback) rain drop number simulations along south-to-north and east-to-west cross-sections through the middle of the model domain. (c,d) Corresponding significance level of mean rain drop number differences**



using the confidence ratio defined in equation (1) – red areas indicate ratio values greater than unity, i.e., significance at or above the 90% confidence level.  (e,f) Difference in rain cloud drop mass (g kg$^{-1}$) (g,h) Corresponding significance level of mean rain drop mass difference.

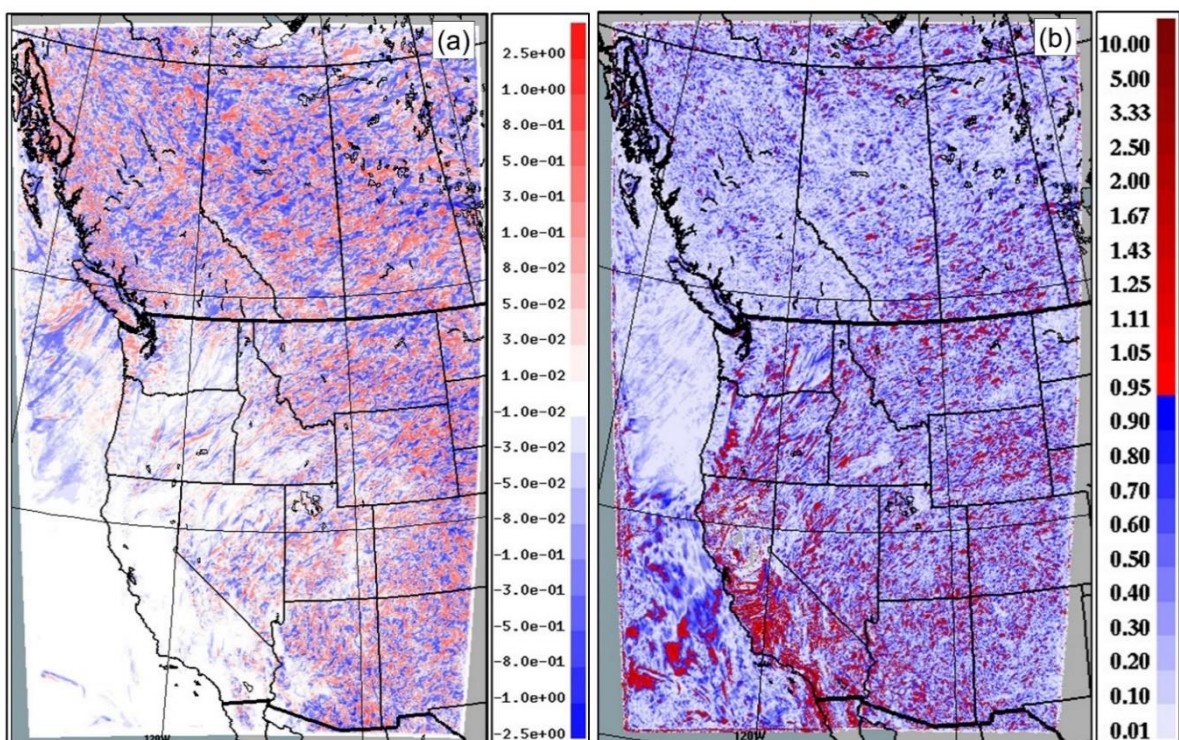

**Figure 18:** (a) Average (Feedback – No Feedback) total surface precipitation during the simulation period.  (b) 90% confidence ratio – values greater than 1 indicate significantly different results at the 90% confidence level.



**Figure 19: Differences in average meteorological fields (feedback – no-feedback; red values indicate more positive values in the feedback simulation than in the no-feedback simulation). Panels show average difference in: (a) specific humidity (g kg$^{-1}$); (b) air temperature (C), (c) dewpoint temperature (C), (d) surface pressure (mb), (e) planetary boundary layer height (m), (f) friction velocity (m s$^{-1}$).**





**Figure 20: 90% confidence ratios, same fields as Figure 19. Values greater than 1 indicate significantly different results at or greater than the 90% confidence level.**



**Figure 21: (a,b)** Difference in mean (Feedback – No-Feedback) temperature simulations along south-to-north and east-to-west cross-sections through the middle of the model domain. **(c,d)** Corresponding confidence ratio of mean temperature differences– red areas indicate ratio values greater than unity, i.e., significance at or above the 90% confidence level. Note the reduction in temperatures between hybrid levels 0.90 to 0.70, similar to the findings of Saponaro *et al.,* (2017).

**Figure 22:** **(a,b,c) Difference (Feedback – No-Feedback) in surface mean PM2.5 (ug m⁻³), NO₂ (ppbv) and O₃ (ppbv), respectively. (d,e,f) Corresponding confidence ratio of mean differences – red areas indicate ratio values greater than unity, i.e., significance at or above the 90% confidence level.**

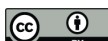


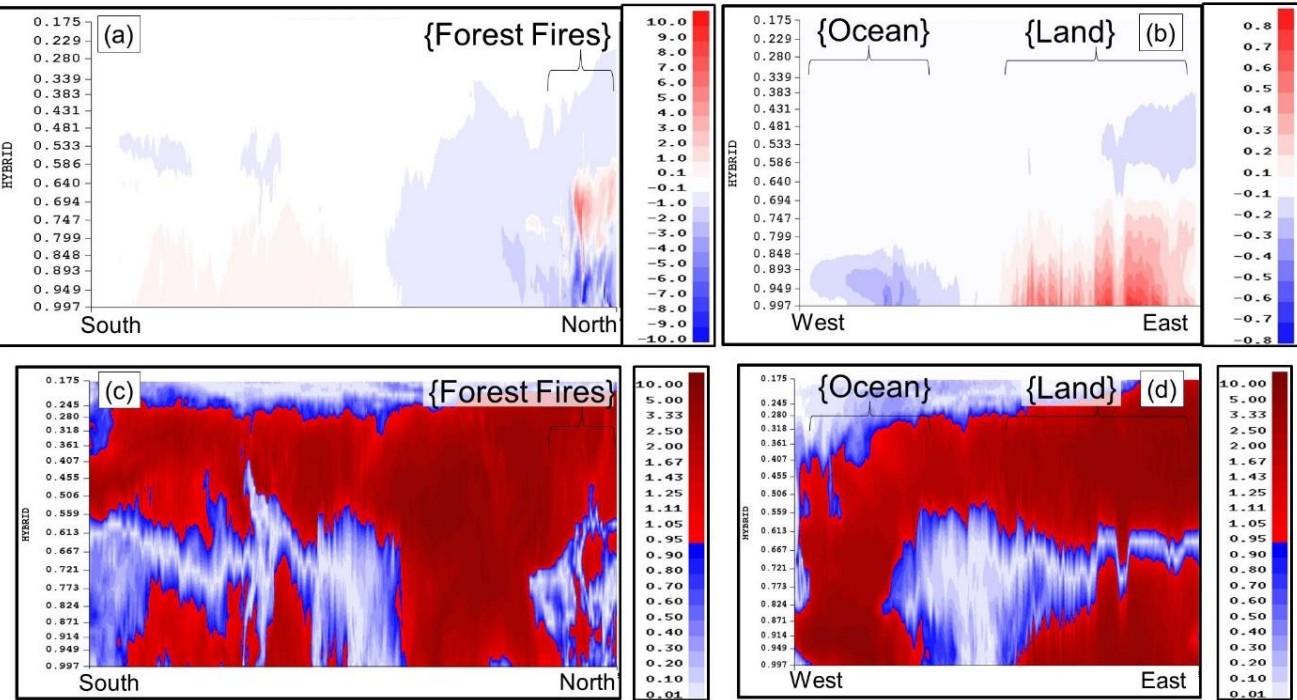

**Figure 23:** (a,b) Difference (Feedback – No-Feedback) in predicted mean PM2.5 (ug m⁻³), along domain-center South-North and West – East cross-sections. (c,d) Corresponding confidence ratio of mean differences – red areas indicate ratio values greater than unity, i.e., significance at or above the 90% confidence level. Note that colour bar scales differ between (a) and (b).



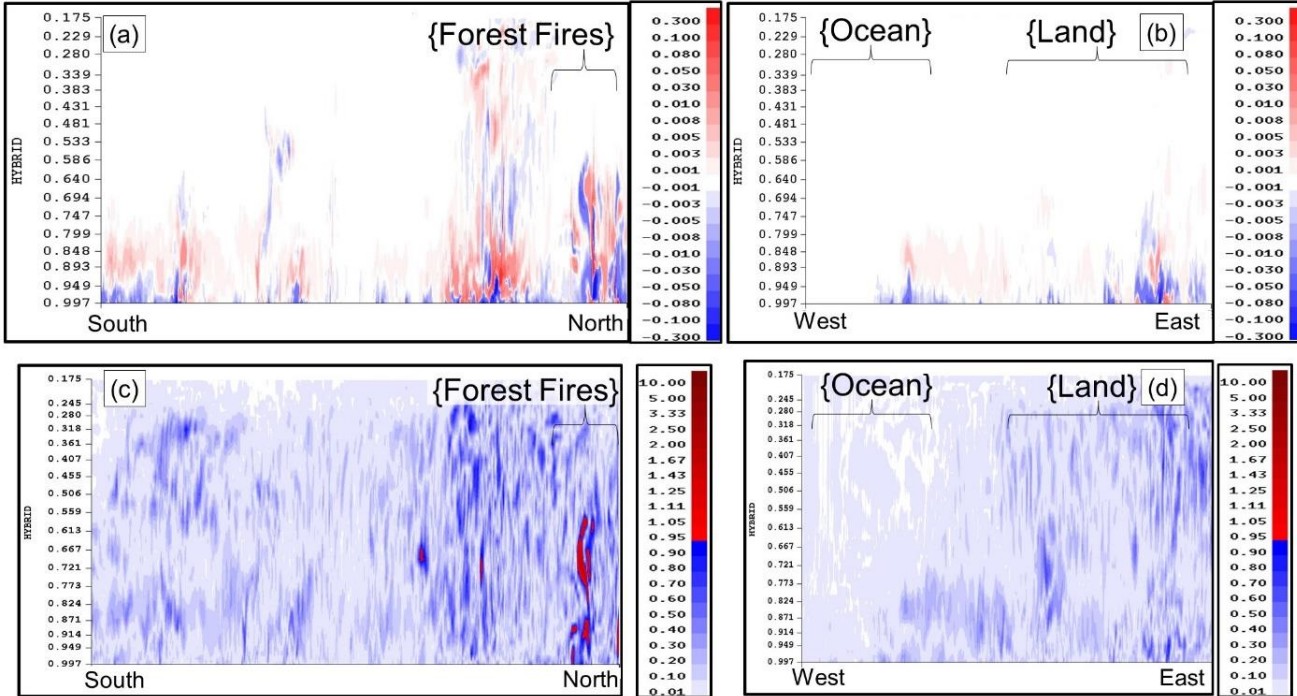

**Figure 24: (a,b) Difference (Feedback – No-Feedback) in predicted mean NO₂ (ppbv), along domain-center South-North and West – East cross-sections. (c,d) Corresponding confidence ratio of mean differences – red areas indicate ratio values greater than unity, i.e., significance at or above the 90% confidence level.**





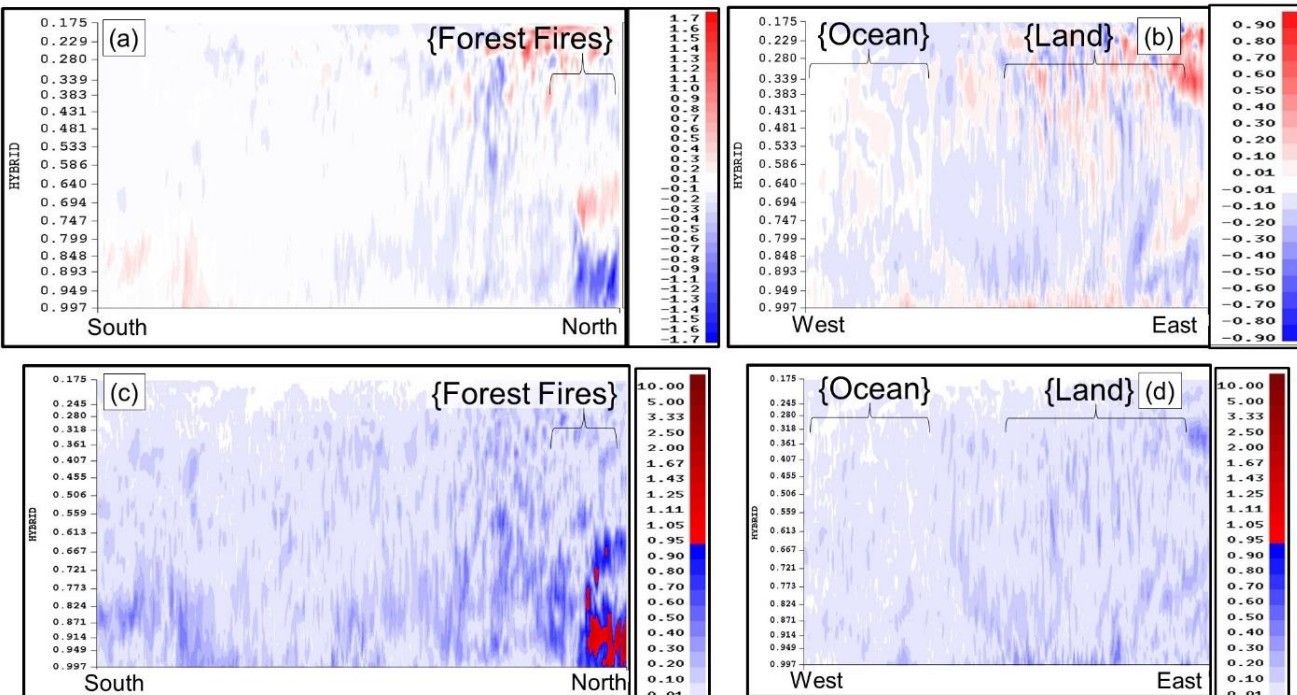

**Figure 25: (a,b) Difference (Feedback – No-Feedback) in predicted mean O₃ (ppbv), along domain-center South-North and West – East cross-sections. (c,d) Corresponding confidence ratio of mean differences – red areas indicate ratio values greater than unity, i.e., significance at or above the 90% confidence level. Note that colour bar scales differ between (a) and (b).**