# Peer review of "Forest Fire Aerosol – Weather Feedbacks over Western North"

_Atmospheric Chemistry and Physics, 2020_

## Referee Comment (RC1) · Anonymous Referee #1 · 13 Nov 2020

This study attempts to evaluate fire-weather feedback processes over Northwestern America using an air quality model. The authors use the GEM-MACH model linked with en experimental configuration of CFEPSv4 as a source of emissions to evaluate how forest emissions impact air quality and weather forecasting performance. The conclusion section suggests that the main objective of the study was to assess if "fully coupled models improve both air-quality and meteorological forecasts". Unfortunately, numerous shortcomings in the deployed methodology make this statement impossible to defend.

First of all, the authors suggest that a fully coupled was used in this study, and try to

differentiate between the fire behavior modeling and coupled air-quality modeling. The authors refer to the very early papers on coupled fire-atmosphere modeling by Clark et al. and Linn et al. that are at that point almost two decades old. Since then, significant progress has been made toward integrating weather-, fire- and air quality models. Fully coupled systems have been in place at least since 2016 (see Kochanski et al. 2016). Unlike the modeling system used in this study, fully coupled models resolve the fundamental interactions between the fire and atmosphere including the plume rise, and the impact of fire released heat and moisture fluxes on local meteorology in line with the chemical transformations of fire emissions. In the context of the fire-atmosphere interactions, the impact of fire heat and moisture fluxes is fundamental and can't be ignored. Multiple papers by Peace at al. 2012, 2015, and 2017 showed that. The presented approach with off-line plume rise calculation neglecting the first-order impact of fires on the atmosphere is not suitable to address the posed question. In fact, the radiative impacts of smoke have been already investigated in the fully coupled framework including resolved fire progression plume rise and chemistry (see Kochanski et al 2019), so the scientific contribution coming from this work due to the use of an overly simplified modeling system is very limited.

The other shortcoming is associated with the lack of proper initialization of the boundary conditions. The authors decided to use two forcing datasets at 10km and 2.5km but did not provide any initialization of the chemical boundary conditions. It is hard to tell if that was the reason for the observed discrepancies between the observed and simulated AOD presented in Figures 14 and 15, but it is evident that the model showed very poor skills in rendering the aerosol optical depth. In the context of that, it is hard to believe in the validity of the presented results and the final message suggesting that fire smoke increased the surface temperature especially when multiple studies published up to date showed something opposite. As a part of the typical smoke shading effect, thick smoke layers tend to decrease incoming solar radiation, induce upper-level warming and low lever colling, increasing atmospheric stability (not decreasing as suggested here). For the discussion of the impact of smoke and aerosols on the boundary layer,

I refer authors to Robock et al 1988, 1991, Lareau & Clements 2015, Yu et al. 2002, Jacobson and Kaufman 2006, and Kochanski et al 2019. The extraordinary findings presented in this study suggesting the opposite, require very strong scientific evidence that has not been provided.

I understand the convenience of using operational forecasts for research purposes, but in this case, a robust modeling setup is needed. If the emissions located outside of the computation domain are believed to be important chemical boundary conditions from a global chemical transport model should be used. Also in the light of uncertainties associated with the emission factors, the mixed results presented in the study do not convincingly present the linked simulations as superior to the uncoupled (unlinked) ones. As indicated by the authors the vertical plume distribution is critical in the context of smoke dispersion but also the radiative impacts. A proper plume rise validation should be one of the first steps in this analysis and could shed some light on the reasons for model efficiency in resolving the AOD.

References:

M Peace, G Mills, (2012) A case study of the 2007 Kangaroo Island bushfires. Journal of Wildland Fire 19, 427-448

M Peace, T Mattner, G Mills, J Kepert, L McCaw (2015), Fire-modified meteorology in a coupled fire–atmosphere model, Journal of Applied Meteorology and Climatology 54 (3), 704-720

M Peace, L Mccaw, B Santos, JD Kepert, N Burrows, RJB Fawcett (2017) Meteorological drivers of extreme fire behaviour during the Waroona bushfire, Western Australia, January 2016. Journal of Southern Hemisphere Earth Systems Science 67 (2), 79-106

Kochanski, A. K., Jenkins, M. A., Yedinak, K., Mandel, J., Beezley, J., & Lamb, B. (2016). Toward an integrated system for fire, smoke, and air quality simulations. International Journal of Wildland Fire, 25(5), 534–568. https://doi.org/10.1071/wf14074

**ACPD**

Interactive
comment

[Figure]

Kochanski, A. K., Mallia, D. V., Fearon, M. G., Mandel, J., Souri, A. H., and Brown, T.: Modeling Wildfire Smoke Feedback Mechanisms Using a Coupled Fire-Atmosphere Model With a Radiatively Active Aerosol Scheme, Journal of Geophysical Research: Atmospheres, 124, 9099–9116, 2019.

Lareau, N. P., & Clements, C. B. (2015). Cold Smoke: Smoke‐induced density currents cause unexpected smoke transport near large wildfires. Atmospheric Chemistry and Physics, 15(20), 11,513–11,520. https://doi.org/10.5194/acp‐15‐11513‐2015

Robock, A. (1988). Enhancement of surface cooling due to forest fire smoke. Science, 242(4880), 911–913. https://doi.org/10.1126/ science.242.4880.911

Robock, A. (1991). Surface cooling due to forest fire smoke. Journal of Geophysical Research, 96(D11), 20869. https://doi.org/10.1029/ 91jd02043

Yu, P., Toon, O. B., Bardeen, C. G., Bucholtz, A., Rosenlof, K. H., Saide, P. E., et al. (2016). Surface dimming by the 2013 Rim Fire simulated by a sectional aerosol model. Journal of Geophysical Research: Atmospheres, 121, 7079–7087. https://doi.org/10.1002/2015JD024702

Yu, H., Liu, S. C., & Dickinson, R. E. (2002). Radiative effects of aerosols on the evolution of the atmospheric boundary layer. Journal of Geophysical Research, 107(D12). https://doi.org/10.1029/2001JD000754

Jacobson, M. Z., & Kaufman, Y. J. (2006). Wind reduction by aerosol particles. Geophysical Research Letters, 33, L24814. https://doi.org/ 10.1029/2006GL027838

---

## Referee Comment (RC2) · Anonymous Referee #2 · 29 Dec 2020

The authors performed a detailed analysis of one-month long period of July 2019 over a part of the US and Canada. An online coupled model GEM-MACH was run with and without aerosol-clouds feedbacks and the difference in its performance was analyzed with attention paid to the regions and the episodes of vegetation fires. As a result of the analysis, the authors declare clear-cut advantages of the coupled meteorology-chemistry forecasts over non-coupled ones in case of non-trivial conditions, such as biomass burning events.

General comments

The discussion on advantages and disadvantages of online coupled models is inter-

esting and important. Being inevitable e.g. in climate- or some episode analysis, the online coupled systems have harder time in other applications, especially in routine operations, such as weather and air quality forecasting. They face the usual set of concerns: Are the resources needed for running such systems on a routine basis justified by the gain? Can these resources be invested in e.g. model resolution, domain size, comprehensiveness of dynamic and chemistry schemes, with better results? The current paper tries to answer some of these questions by applying the GEM-MACH coupled system in forecasting mode with related technicalities and constraints. In that sense, I found the paper undoubtedly interesting.

The general problem, however, was that the declared outcome of the analysis does not follow from the material. The authors state: "incorporating aerosol direct and indirect effect feedbacks can significantly improve the accuracy of weather and air quality forecasts". I struggled to find ground for it.

The implementation of the forecasts has several compromises, which seem to have more than enough power to overshadow any effect of the system complexity. Arguably the most-significant problem is the strange decision to use a decade-old MACC reanalysis as the boundary conditions for the run. With all efforts, I could not understand it: the domain is comparatively small, boundaries are important and the Copernicus operational forecast is available from the same ECMWF source. It covers more species than the old MACC reanalysis, embeds quite detailed fire data, involves satellite data assimilation and has better resolution. One can also look at ICAP ensemble of global aerosol and atmospheric composition models: forecasts of some of them are available. The list can be extended. There is no shortage of real-time data and forecasts, many easily available, why not to use them? The extra effort is a blip compare to other arrangements.

A possible result of the inadequate boundary conditions was a very large bias of AOD – up to 0.25-0.3 in the Figure 14, which constitutes almost an order of magnitude. Comparing to that error, the effect of coupling is negligible. The problem is noted by the

authors but with no follow-up. However, if the missing aerosols were indeed from the boundaries and consist of reactive and soluble particles of fire smoke or sea salt, the chemical, aerosol, and cloud processes of the simulations are completely jeopardized, and no conclusions can be drawn. This suspicion is supported by the low correlation coefficient for PM2.5 (< 0.3, Table 2), which also suggests serious deficiency in the aerosol content and processes.

A general expectation from incorporation of new important processes is that it must lead to a better system behavior, ability to follow the changes in the environment and, consequently, to better correlation with observations. Unfortunately, this crucial statistical parameter did not show any difference between the runs. An exception is the PM2.5 score in Western Canada where the no-feedback run won (table 2), which essentially disproves the paper conclusions. Improvements due to coupling were noticeable only for bias and statistics related to it. But with no effect on correlation, the same or even more significant effect could be achieved, apart from boundary conditions, by a trivial bias correction, either in the aerosol formation/removal schemes or even as post-processing.

A similar question arises from the fire plume coupling. Appreciating the idea and efforts, I could not miss the remark that the approach does not account for the heat released by the fires. Being usually a reasonable compromise between the complexity and gain, in this case it is hardly correct. The model takes a great deal of efforts to account for the aerosol impacts on energy budget but this add-on can easily appear smaller than the neglected impact of fires.

I also noticed seemingly unclear / contradicting sentences concerning the coupling: a statement in line 258 probably means that the P3 was used for the AIE whereas the explanation in line 243 says that P3 uses prescribed particles features rather than the data from the aerosol module. So, was the coupling so full as the paper repeatedly says?

Presentation of the material is heavy. The paper is monumental and wordy, in many places more resembling a textbook than a focused research manuscript. It pays off to a devoted reader but sometimes, this approach backfires. For instance, a long description of the simulations leaves out many important features of the setup and takes a great effort from a reader to grasp it. A summary table is needed here.

Summarizing, I found the paper heavy but interesting to read, presenting a good outline of the state of art and contributing to the discussion on added value of the online-coupled models. However, its conclusions do not follow from the presented material, which rather shows almost the opposite. As a result, a somewhat pushy bold style of the presentation does not look convincing and eventually annoys the reader.

I would suggest a major revision of the manuscript turning it into a discussion review paper. It should present the experiment in a neutral way and discuss its features, contributions from different system components to its overall skills, as well as the ways for making it better, both via the feedback mechanisms and via simpler steps of improving the models themselves and the setup of the simulations. In such a form, the extensive text will become an asset.

---

## Author Comment (AC1) · 16 Mar 2021

Please note: we have attached a PDF version of this response as a supplemental file, which may have better formatting, for the Referee's convenience.

Response to Referees, "Forest Fire Aerosol – Weather Feedbacks over Western North America Using a High-Resolution, On-line Coupled, Air-Quality Model".

Anonymous Referee #1:

We thank the anonymous referee for their review of our paper, though we beg to differ
with their conclusion. In their review, they state that our main objective to assess if "fully coupled models improve both air-quality and meteorological forecasts" has "numerous shortcomings in the deployed methodology make this statement impossible to defend." We disagree with the referee's conclusion, as we outline below.

Re: "First of all, the authors suggest that a fully coupled was used in this study, and try to differentiate between the fire behavior modeling and coupled air-quality modeling".

Response: The referee challenges whether "a fully coupled was used in this study" and we acknowledge the use of the term "fully coupled" may mean different things to different sub-fields within forecasting modelling groups. The model presented in our manuscript couples in the context of a continental-scale operational forecast, specifically, the large scale processes of the aerosol direct and indirect effects (resulting of emissions from fires and other sources) and fire behaviour (specifically fuel consumption, fire intensity) as opposed to a coupling with fire spread and growth on the landscape. We made an effort to recognize and distinguish our use of the term "coupling" from other models that coupled meteorology to fire behaviour and growth on a local-scale landscape. We referenced Clark et al. and Linn et al. specifically as they were the seminal works on this topic. The referee suggests using more current references and we shall accommodate. We have also replaced "fully coupled" with "on-line coupled", referencing and following the definition given in Galmarini et al. (2015), and corrected Figure 3 to be more specific.

The referee referenced works using WRFSfire. We acknowledge and applaud such research efforts but wish to remind the referee of the context of our work, in terms of a much larger regional scale application, and of operational forecast constraints on the complexity of modelling fire emissions, chemistry and weather, as well as the scale-dependence of fire growth modelling. Kochanski et al 2016 documents simulations of two wildfires: the first with 4 nested domains with an inner domain with a resolution of 500m; the second with 5 domains and a inner 444m aresolution. Kochanski et al. 2019 describes a high-resolution simulation of six fires using 3 nested domains captured on

the same 1.33 km resolution inner domain. Peace et al. 2015 use WRFSfire on a 2007 Australian fire using 4 nested domains with a 22 km x 22km inner domain having an even finer grid spacing of 222 m (we are uncertain why the referee referenced Peace et al. 2012, 2017 as no coupled NWP-fire modeling is involved). The key point here is that all of these simulations are intended to allow the detailed study of the local scale of domains covering individual fires. They are not operational forecast systems capable of predicting the wider scale impacts of multi-source forest fire emissions on the aerosol direct and indirect effects over a large region. When going to the very local scale in the cited papers, additional parameterization detail is indeed warranted – however, at the current capacity of supercomputers, models of the nature described by the references are limited in size and scale, and still require a large amount of processing time – in the case of Kochanski et al. 2016, simulations took 13 to 30 hours depending on configurations. That time estimate likely does not account for the time spent collecting, preparation, and preprocessing the necessary data, and is prohibitively large for regional scale forecasting.

What we have presented in contrast in our manuscript were the results of simulations "carried out for the period 4 July through 5 August 2019, at 2.5-km horizontal grid cell size, over a 2250 x 3425 km2 domain covering western Canada and USA, . . . as part of the FIREX-AQ ensemble forecast." These regional scale forecast simulations were required to fit within a 3 hour operational forecast window; all processing had to take place within a three hour hard limit on the processing time for product delivery. Essentially the referee is criticizing our operational, synoptic-scale model for not fully including an set of high-resolution micro-scale processes which, while giving an excellent idea of what happens on a local scale, can not be run in an operational, regional-scale forecast, context.

Regarding the referee's suggestion that a "proper plume rise validation should be one of the first steps in this analysis and could shed some light on the reasons for model efficiency in resolving the AOD." This indeed was done by Griffin et al. (2019) using the

same modelling framework as described in our work – in that paper (reference below) modelled plume heights similar (albeit slightly higher) to those observed by satellite observation. This reference was overlooked in the submitted work but is included in the revised manuscript.

To address these issues, and to place both our own work and that referenced by the reviewer in context, we have modified the paper the Introduction as follows:

"The latter point is worthy of note in the context of the direct and indirect feedback studies noted above – both climate and weather simulations with prescribed forest fire emissions have consistently resulted in large perturbations of weather patterns in the vicinity of the forest fires. However, their approaches for predicting forest fire plume rise and fire intensity and fuel consumption in operational regional scale forecasts up until now have relied on weather forecast information provided a priori and hence lacking those meteorological feedback effects. The connection of the ADE and AIE within a regional air-quality and weather forecast model context is referred to as "coupling", with such a model being described in that body of literature as "fully-coupled", or "aerosol-aware". However, several researchers have examined aerosol-radiative coupling along with fire spread and growth (as opposed to fire intensity and fuel consumption). The latter work employs very high-resolution forest fire spread and growth models, and due to their very high resolution, an additional level of coupling, that of interaction of dynamic meteorology with the heat released by the fire, may be included. However, the resolution requirements for these models (and their need for a relatively small computational time step) constrains their application to a relatively small region. A requirement for these approaches is the use of a very high resolution fire growth model imbedded within the air-quality model. At these resolutions, the simulated local-scale meteorology determines fire spread on the landscape, which in modifies the temperature and wind fields, in turn affecting future fire spread. The seminal work on this topic was carried out by Clark et al. (1996), and Linn et al. (2002). More recent work includes the development of WRF-SFIRE model (Mandel et al., 2011; Coen et al., 2013) ), with

full chemistry added in the WRFSC model (Kochanski et al., 2016). Examples of the resolution required for these models include inner domain resolutions of 444 m with an imbedded fire model mesh of 22.2 m resolution, and a time step of 3.3 seconds (Kochanski et al., 2016); 1.33 km with an imbedded fire model mesh of 67.7m, and a time step of 2 seconds (Kochanski et al., 2019), and 222m, with a fire model mesh of 22m and a time step of 2 seconds (Peace et al., 2015). Kochanski et al (2016) also noted a 13 to 30 hour computational time requirement to run their high-resolution modelling system. These modelling efforts allow for this additional level of coupling – but at the expense of additional computation time preventing, at the current state of supercomputer processing, their application on synoptic-scale forecast domains combined with a full gas chemistry and size-resolved multi-component particle chemistry representation. Here we explore the effects of fire emissions characterized by fire intensity and fuel consumption modelling on the aerosol direct and indirect effects over synoptic scale domain. Our coupling refers to that between the aerosols released by fires and other sources to meteorology through the ADE and AIE, with the resulting changes in meteorology in turn influencing fire fire intensity and fuel consumption, in turn influencing emissions height and distribution, closing this feedback loop. We do not implement a very high resolution fire growth model, noting that this is impractical for operational forecasts at the current time, while showing that synoptic scale 2.5km simulations incorporating fire feedbacks may be carried out within an operational window with currently available supercomputers. As shown below, we find that a sufficiently substantial feedback between the aerosol direct and indirect effects can be discerned to change the vertical distribution of emitted pollutants."

With regards to plume rise validation, we have included the following addition to the section describing the CFFEPS model: "A recent evaluation of the plume heights predicted by CFFEPS was carried out utilizing MISR and TROPOMI satellite retrieval data (Griffin et al, 2020). Seventy cases studied using MISR data showed good agreement between satellite and CFFEPS-predicted maximum and mean plume heights (maximum plume height observed versus predicted values and standard deviations: $1.7\pm0.9$

versus 2.0±1.0 km; mean plume height observed versus predicted: 1.3±0.6 versus 1.3±0.4 km). 671 cases studied using TROPOMI data also showed a reasonable agreement, with CFFEPS showing a small tendency to overpredict heights (maximum observed versus predicted plume heights 2.2±1.6 versus 2.5±1.2 km; mean observed versus predicted plume heights 0.7±0.5 versus 1.1±0.6 km). "

Re: "lack of proper initialization of the boundary conditions".

"The authors decided to use two forcing datasets...but did not provide any initialization of the chemical boundary conditions."

Response: this is not quite correct: we made use of a several week spin-up to allow the simulation to reach a local steady state. Also, we made use of a chemical lateral boundary condition based on a 2009 MOZART climatology. We were uncertain whether the referee's comment was intended to suggest that no boundary values were used (or zero values were used) for the chemical initial and boundary conditions – this was not the case.

"It is hard to tell if that was the reason for the observed discrepancies between the observed and simulated AOD presented in Figures 14 and 15, but it is evident that the model showed very poor skill in rendering the aerosol optical depth."

We note that AOD is not directly predicted by the model. The model predicts aerosols and their physical and chemical properties; AOD is then calculated/evaluated based on the modelled aerosols and atmospheric conditions. There are different ways to calculate/evaluate AOD (as referenced in the submitted manuscript, Curci et al 2015 reference), which introduces significant uncertainty in model evaluated AOD. At the same time, the model showed a relatively high skill in predictions of PM2.5, NO2, and O3, as well as showing that the feedback implementation improved both the forecasts of meteorology and temperature. The reviewer focussed on a single parameter where the performance was low.

"Final messaging suggesting that the fire smoke increased the surface temperature especially when multiple studies to date showed something opposite".

The Referee's interpretation is not correct – our submitted work did show a decrease in temperatures in the troposphere between roughly 950 and 720 mb (original Figure 21(a), now Figure 20(a)) associated with the feedbacks, which reached the ground in the region most affected by the fires. This can also be seen in original Figure 19(b), now Figure 18(b), where small regions of decreased surface temperature can be discerned. In the southern part of the domain, the temperature increases were more prominent, due to this fire-smoke-free region having, in the feedback simulation, less aerosol impacts relative to the aerosol climatological properties used in the no-feedback simulation. This is an important point – which we made several times in the original manuscript – our "no-feedback" simulations are not "no aerosol" simulations. Consequently the impacts of forest fire smoke – relative to that baseline – will be decreased in comparison to a simulation in which there are no aerosol optical properties whatsoever. The effect of smoke to reduce temperatures at the surface is something we are well aware of from our own past work as well (e.g. Makar et al., 2015(a)) reference in our original manuscript, where we showed a substantial decrease in surface temperatures associated with large fires in Russia in 2010.

Despite the above points, we felt that an investigation of the impact of the chemical lateral boundary conditions was warranted, given that both reviewers raised the issue. We had also noted the potential for chemical lateral boundary conditions to be a source of uncertainty in our submitted paper, and we felt that this was worth investigating. Examining satellite AOD data for the northern hemisphere, we noted the presence of high AOD events associated with sources in Asia and Siberia (with the latter from forest fires) crossing the Bering Strait – an upwind boundary condition potentially of concern, in addition to the Alaska fires we mentioned in our original manuscript. More importantly, our other reviewer also pointed us to a publicly available source of global chemical reanalysis data, which recently became available from ECMWF, in the

same year our operational forecasts for FIREX-AQ took place (Innes et al., 2019). In a three-month effort following the receipt of the reviews for our original submission, we obtained global ECMWF reanalyses for our forecast period, converted these to our model grid and speciation, and re-ran the feedback and no-feedback simulations, this time making use of spatiotemporal varying ECMWF reanalysis data to provide boundary conditions for a 10-km resolution GEM-MACH simulation, which in turn was used to provide chemical lateral boundary conditions for the inner 2.5km GEM-MACH domain. We regenerated all Figures and analyses, and compared the results to our previous simulations. These simulations have been incorporated into our revised manuscript, with the original simulations being mentioned therein as a reference to the Discussion paper. Some of our main findings (in the revised manuscript) as follows:

- The impact of direct and indirect feedbacks on both weather and air-quality forecast performance is for the most part unchanged. A better forecast was found when the feedbacks were incorporated. That is, this key finding from our original work was not influenced by the boundary conditions used previously.

- The use of the chemical lateral boundary conditions did, however, improve the model's performance for AOD, with the slope of the observation (y) to model (x) line going from 0.15 to 0.56. That is, the AOD underprediction was improved by a factor of 3.7.

- The higher aerosol loading from the chemical lateral boundary conditions increased the spatial extent of the region with lower surface temperatures such that most of the northern portion of the domain now experienced lower temperatures resulting from the feedbacks. We also expanded the discussion of stability in the model based on these findings – near surface stability below the smoke layer decreased over a larger region, while the stability above the smoke layer increased.

- However, relative to the original simulation employing a MOZART climatology along the lateral boundary, most of the performance metrics for air-quality became worse, using the new chemical lateral boundary condition. In particular, PM2.5, O3 and NO2

biases, which had varied between small positive to small negative, increased in magnitude and were uniformly positive. So, while this choice of chemical lateral boundary conditions improved AOD, it did so at the expense of increasing PM2.5 surface biases (and degrading most air-quality metrics). The feedbacks nevertheless still resulted in improved forecasts relative to the no-feedback simulations. We therefore conclude in our revised manuscript that the chemical lateral boundary conditions have a key impact on model performance – and that improving those boundary conditions should be a focus for future work.

Referee #2:

We thank the Referee for the comments, particularly for the Referee's suggestion of making use of ECMWF boundary conditions.

Re: "Arguably the most-significant problem is the strange decision to use a decade-old MACC reanalysis as the boundary conditions for the run. With all efforts, I could not understand it: the domain is comparatively small, boundaries are important and the Copernicus operational forecast is available from the same ECMWF source. It covers more species than the old MACC reanalysis, embeds quite detailed fire data, involves satellite data assimilation and has better resolution. One can also look at ICAP ensemble of global aerosol and atmospheric composition models: forecasts of some of them are available. The list can be extended. There is no shortage of real-time data and forecasts, many easily available, why not to use them? The extra effort is a blip compare to other arrangements."

We agree; this was a good point. We were constrained at the time of the forecasts to an operational time window, and in the real-time forecasting used for our submitted work, we had not anticipated impacts from trans-Pacific transport. Our hope was that the large size of the domain would preclude the need for a time dependent larger domain boundary condition (e.g. from our own modelling system, let alone a global model). In order to follow up on the Referee's suggestion, we first examined the available satellite data during our simulation period to determine the extent to which North American, versus Asian, upwind sources could be affecting our model domain. Would a North American domain simulation starting from our original MOZART climatology be sufficient? In this examination, we were able to spot several events of high AOD crossing the Bering Strait from Russia and Asia, with fires in Siberia being the likely source of the high AOD based on time-coincident hotspot analysis. From this work, we decided that we should make use of ECMWF 3 hour reanalysis data (given that the referee recommended this model) to provide chemical lateral boundary conditions to a 10 km resolution North American domain GEM-MACH simulation, which in turn was used to provide boundary conditions to the inner 2.5km GEM-MACH western North America domain employed in our initial submission to ACPD. That is, in addition to the use of ECMWF boundary conditions at the North American scale, we have added a GEM-MACH simulation at the continental scale, with 10km resolution, to provide the emissions and chemical processing for the remainder of North America: the result is a cascade of lateral boundary condition (ECMWF reanalysis boundary conditions used by the North American GEM-MACH simulation, then North American GEM-MACH simulation provided boundary conditions for the inner 2.5km domain..

We note, however, that the Referee's comment that "The extra effort is a blip compare to other arrangements" underestimates the level of effort required to make use of another modelling system's data as chemical lateral boundary conditions: we did make use of ECMWF's reanalysis fields for our revised manuscript, and we are grateful for the availability of ECMWF's global model 3 hour reanalysis fields, but obtaining those fields, converting them to a form which could be used in our model (including dealing with differences in chemical speciation and particle size resolution between their modelling system and ours), and repeating the feedback and no-feedback simulations took 2.5 months to carry out including the setup time, though the simulations themselves run at 1/8 real time. This was not a "blip" in effort, as described by the Referee. Nevertheless, it was definitely worth "checking out".

All analyses and figures associated with our original submission to ACPD were repeated, with this new configuration. Our main findings from the new analysis include:

With the new configuration, the feedback forecast still outperformed the no-feedback forecast, for both meteorological and chemical variables. That is, despite a fairly substantial change in the boundary conditions, most of our conclusions from the original paper remain unchanged. Weather forecast scores matched or were improved at the 90% confidence level, and 35 out of 48 of the PM2.5, NO2 and OÂň3 statistical comparisons were better with the feedback forecast. We feel that this addresses the Referee's concern that "if the missing aerosols were indeed from the boundaries and consist of reactive and soluble particles of fire smoke or sea salt, the chemical, aerosol and cloud processes of the simulations are completely jeopardized, and no conclusions can be drawn." On the contrary – having incorporated the boundary conditions from a global chemical model's reanalysis, our most significant conclusion, that feedbacks provide a small but significant improvement in both weather and chemical forecast quality, remains unchanged.

The use of the revised chemical lateral boundary conditions greatly improved the model's performance for AOD, with the slope of the observation (y) to model (x) line going from 0.15 to 0.56, a factor 3.7x improvement. Our revised Figure 14, where we compare all of the satellite overpass AOD pixels during our simulation time to the equivalent model AOD values. We feel that this addresses the Referee's comment, "A possible result of the inadequate boundary conditions was a very large bias of AOD – up to 0.25 to 0.3 in the Figure 14, which constitutes almost an order of magnitude. Comparing to that error, the effect of coupling is negligible."

The higher aerosol loading from the chemical lateral boundary conditions increased the spatial extent of the region with lower surface temperatures such that most of the northern portion of the domain now experienced lower temperatures resulting from the feedbacks, rather than just small areas in the northern part of the 2.5km domain. We also expanded the discussion of stability in the model based on these findings –

near surface stability below the smoke layer decreased over a larger region, while the stability above the smoke layer increased.

We also saw a trend towards further improvements in the temperature profile predictions associated with feedbacks, particularly for the 12 hour forecast values; a greater proportion of the vertical column now had temperature improvements at greater than the 90% confidence level than with the previous simulation.

However, relative to the original simulation employing a MOZART climatology along the lateral boundary, most of the performance metrics (for both feedback and no-feedback forecasts) for air-quality became worse with the use of the ECMWF and 10km GEM-MACH simulations as boundary conditions. In particular, PM2.5, O3 and NO2 biases, which had varied between small positive to small negative, increased in magnitude and were uniformly positive. So, while this choice of chemical lateral boundary conditions improved AOD, it did so at the expense of increasing PM2.5 surface biases (and degrading other air-quality metrics). We therefore conclude in our revised manuscript that the chemical lateral boundary conditions do in fact have a key impact on model performance – and that improving those boundary conditions should be a focus for future work.

Re: "A general expectation from incorporation of new processes is that it must lead to a better system behavior, ability to follow the changes in the environment, and, consequently to better correlation with observations. Unfortunately this crucial parameter did not show any difference between the runs….But with no effect on correlation, the same or more significant effect could be achieved, by a trivial bias correction, either in the aerosol formation/removal schemes or even as post-processing."

We were surprised by the Referee's focus on correlation coefficient as a key metric to determine model performance, and in our revised manuscript we include references in the literature for the past few decades describing its benefits versus limitations for model performance analysis, as well as for our other performance metrics. Our intent with the variety of metrics used for model evaluation was to provide a wide range of performance metrics, given that each has its strengths and weaknesses. The Referee's statements on correlation coefficient goes against recent literature on model performance evaluation, which also stress the need to evaluate models against multiple metrics. For example, guidance available from the ECMWF's Copernicus Atmosphere Monitoring Service (http://macc-raq-op.meteo.fr/doc/USER_GUIDE_VERIFICATION_STATISTICS.pdf) makes use of mean bias, modified mean bias, root mean square error, fractional gross error, as well as correlation, for model evaluation, noting that RMSE (not correlation coefficient) "is the most common estimator of the accuracy of forecasts". Correlation coefficient, while commonly used for model evaluation, has been identified as being unable to distinguish systematic model underestimation, which is better captured by other metrics (Yu et al., 2006). Other potential pitfalls of the correlation coefficient include outliers providing a false sense of relationship, no being information on whether the data series have a similar magnitude (Duveiller et al., 2016), and clusters of model-observation pairs providing a false sense of relationship (Aggarwal and Ranganathan, 2016). Krause et al. (2005) recommended "r2 alone should not be used for model quantification, because it can produce high values for very bad model results".

We agree that a post-processing bias correction could be performed on a model's output, but that is not relevant to the use of bias as an a priori model performance metric.

We realized based on the Referee's comment that our original manuscript did not include a summary of the above information, and did not provide a description of the pro's and con's of the different model performance metrics used in our work for air-quality model evaluation. We have added a paragraph to our original text to address this issue, which places the air-quality model performance metrics we have used in the context of past work on the subject of model evaluation. We have also added a more detailed description of each performance metric, along with their advantages and disadvantages and additional references, in a Supplemental Information Appendix to our revised manuscript. The additional paragraph to the main body of the text follows:

"Improvements to air quality model performance metrics have been a focus for research since the 1980's starting with dispersion model evaluation (Fox, 1981), and the identification of mean bias and normalized mean square error as potentially useful metrics to complement the Pearson correlation coefficient (Hanna, 1988). More recently, the Pearson correlation coefficient has been noted as being capably of producing high values for relatively poor model results (Krause et al., 2005), as well as being unable to distinguish systematic model underestimation (Yu et al., 2006), unable to provide information on whether data series have a similar magnitude and capable of providing a false sense of relationship where none exists due to outliers (Duveiller et al., 2016) and clusters of model-observation pairs (Aggarwal and Ranganathan, 2016). More recently, model evaluation has focused on metrics which do not have the tendency to weight the higher magnitude values unduly (a particularly useful property with air-quality variables which may vary by several orders of magnitude), which are dimensionless (allowing a comparison across different evaluated variables), and which are bounded and symmetric (properties allowing comparisons to be made and equally valued across the entire range of possible concentrations; e.g. Yu et al. (2006)). Metrics such as the modified coefficient of efficiency (Legates and McCabe, 1999) and the more recent incarnations of the Index Of Agreement (Willmott et al., 2012) are examples of the more recent metrics used for air-quality model evaluation. Here, we have made use of a range of metrics, spanning the literature on this topic, with the understanding that the properties of different metrics vary, that no single metric provides a perfect means of evaluating model performance, and that a variety of metrics should be applied. The metrics used here span the variety that have appeared in the literature since the early 1980's, and include Factor of 2, Mean Bias, Mean Gross Error, Normalized Mean Gross Error, Correlation Coefficient, Root Mean Square Error, Coefficient of Efficiency, and Index of Agreement. The formulae for these metrics and a brief description of their relative advantages and disadvantages appears in Appendix A (Supplemental Information)."

Re: "I could not miss the remark that the approach does not account for the heat released by fires."

This sentence could have been better worded. The plume rise approach does in fact account for the heat released by fires; the thermodynamics of plume rise are simulated using CFFEPS4.0 as a sub-gridscale process. The intent here was to draw a distinction between the scale of our model and the very high resolution, subgridscale fire growth models such as WRFSC. We have modified the text in two places to better explain this point, specifically:

The original text:

"However, we note that the local scale weather modifications due to the addition of forest fire heat to the atmosphere are not yet incorporated into fire spread or GEM microphysics. Specifically, when the feedback version of GEM-MACH incorporating CFFEPSv4.0 is used in its fully coupled configuration, CFFEPSv4.0 calculates forest fire plume rise using the meteorological predictions which include the ADE and AIE generated by forest fire aerosols on atmospheric stability from the current fully-coupled model timestep."

Has been changed to:

"However, we note that CFFEPSv4.0 employs forest fire heat to determine plume rise as a subgridscale thermodynamic process parameterization rather than a very high resolution explicit fire growth parameterization; the very local scale weather modifications due to the addition of forest fire heat to the atmosphere are not incorporated into fire spread or GEM microphysics. Specifically, when the feedback version of GEM-MACH incorporating CFFEPSv4.0 is used in a coupled configuration, CFFEPSv4.0 uses estimates of the heat released to calculate forest fire plume rise. These calculations employ lapse rates at the fire locations, that with feedbacks enabled, include the ADE and AIE generated by forest fire aerosols on atmospheric stability within the current coupled model timestep. This is in contrast to earlier off-line implementations

of CFFEPS, which made use of a priori non-feedback weather forecast lapse rates. "

Re: "I also noticed seemingly unclear / contradicting sentences concerning the coupling: a statement in line 258 probably means that the P3 was used for the AIE whereas the explanation in line 243 says that P3 uses prescribed particles features rather than the data from the aerosol module. So, was the coupling so full as the paper repeatedly says?"

We've clarified the paper to better explain this point. The feedback option uses on-line two-way coupling; the no-feedback version does not have this coupling. Rather, the no-feedback version of the code makes use of climatological aerosol optical and CCN properties. We had made this point elsewhere in the original manuscript, but we've repeated it again, this time in the context of the table the Referee requested to reduce the paper size.

The previous two sentences at line 243 were:

"This droplet nucleation replaces the decoupled model's existing droplet nucleation calculation in the Predicted Particle Properties (P3) microphysics scheme (Morrison and Milbrandt, 2015, Milbrandt and Morrison, 2016). The latter assumes an invariant aerosol population of a single lognormal size distribution (with a geometric mean diameter of 100 nm and total aerosol number concentration of 300 cm-3, consisting of pure ammonium sulphate; Morrison and Grabowski, 2008)."

The original line 258 was:

"the AIE parameterization was modified for use with the P3 cloud microphysics scheme"

Following the Referee's recommendation regarding "heaviness" of the paper, this has been replaced by the following rows in the new Table 1 in the revised manuscript, which includes a row with the following entries in each column:

Process: Aerosol Indirect Effect

Description: Feedback simulations: Modified P3 cloud microphysics scheme, driven by an aerosol size and speciation specific nucleation scheme (Abdul-Razzak and Ghan, 2002). No-feedback implementation: P3 scheme driven by an invariant aerosol population of a single lognormal size distribution (with a geometric mean diameter of 100 nm and total aerosol number of 300 cm-3 consisting of pure ammonium sulphate). The prognostic cloud droplet number and mass mixing ratios from the P3 microphysics are then transferred back to the chemistry module for using in cloud processing of gases and aerosols (cloud scavenging and chemistry) calculations, completing the AIE feedback process loop in the case of the feedback implementation (Gong et al., 2015).

References:Gong et al. (2015), Abdul-Razzak and Ghan (2002), Morrison and Milbrandt (2015), Milbrandt and Morrison (2016), Morrison and Grabowski (2008).

Re: "Presentation of the material is heavy.... a long description of the simulations leaves out many important features of the setup and takes a greater effort from a reader to grasp it. A summary table is needed here."

As suggested by the reviewer, we have replaced most of our original section 2.1 with the new Table 1, which summarizes the same information. We assume that "the many important features left out of the setup" were encompassed by the Referee's comments above (and we hope that our responses have addressed those concerns). The description of each model performance metric appears in the Supplemental Information in order to not increase the heaviness of the paper.

We have also moved the methodology used to generate 90% confidence intervals to the SI.

Re: "However its conclusions do not follow from the presented material, which rather shows almost the opposite".

We note that our Figures 6, 7, 8, 9, 10, 11, 12, 13 show our feedback weather forecast matching or outperforming the no-feedback weather forecast at greater than 90% confidence at most times and heights in the atmosphere. Our Table 3 shows the feedback air-quality forecast outperforming the no-feedback forecast for 35 out of the 48 metrics being compared.

In order to not risk "over-stating" our results, we have modified the Abstract to be more specific:

The Abstract sentences:

"Incorporating feedbacks resulted in improvements in most metrics of both air-quality and meteorological model forecast performance, through comparison of no-feedback and feedback simulations with surface, radiosonde, and satellite observations. For the meteorological simulations, these improvements occurred at greater than the 90% confidence level."

. . . were modified to read:

"Incorporating feedbacks resulted in weather forecast performance that exceeded or matched the no-feedback forecast, at greater than 90% confidence, at most times and heights in the atmosphere. The feedback forecast out-performed the feedback forecast at 35 out of 48 statistical evaluation scores, for PM2.5, NO2 and O3."

Similarly, the relevant portion of the conclusions has been modified to "Within the high resolution domain size employed here, improvements or matching weather forecast performance was seen for most times and heights in the atmosphere, at greater than 90% confidence. Improvements in model performance for surface PM2.5, NO2 and O3 were also found, across most statistical measures (35 out of 48 statistical evaluation scores showed improvements)."

In our original submission, the first sentence of the Conclusions read:

"The work carried out here suggests that the answers to our two research questions ("Can fully coupled models improve both air-quality and meteorological forecasts?" and "Are the changes in forest fire forecasts associated with implementing forest fire emissions within a fully coupled model sufficient to significantly perturb weather and meteorology?") are both a qualified "yes". "

We have placed the word "qualified" in italics in the modified version – that is, we were already stressing the caveats regarding the improvement in model performance. It is there, and seems to be robust from our repeat simulations using ECMWF boundary conditions – it is a small but nevertheless significant improvement in forecast performance.

In our original submission's section 3.3 (Model Evaluation Summary), we had also noted, "Overall, the incorporation of feedbacks in this study has resulted in improvements in weather and air-quality forecast accuracy, albeit with some caveats."

We have modified the text so that the "albeit with some caveats" appears in italics, for emphasis, and used the phrase "matching performance or improvements at the 90% confidence level."

In our revised version of this section, we include the specific number of improved scores for the model evaluation, "Scores for surface PM2.5, NO2, and O3 also generally improved with the incorporation of feedbacks (35 out of 48 comparisons showed improvements)."

Please also note the supplement to this comment:
https://acp.copernicus.org/preprints/acp-2020-938/acp-2020-938-AC1-supplement.pdf